EMBO
Molecular Medicine

# Pyrin inflammasome-driven erosive arthritis caused by unprenylated RHO GTPase signaling

Murali K Akula [1,2]✉, Elisabeth Gilis[1,3], Pieter Hertens [1,2], Lieselotte Vande Walle[4], Mozes Sze[1,2],
Julie Coudenys[1,3], Yunus Incik[1,2], Omar Khan [5], Martin O Bergo [6], Dirk Elewaut [1,3],
Andy Wullaert [1,4,7], Mohamed Lamkanfi [4] & Geert van Loo [1,2]✉

## Abstract

**Geranylgeranyl pyrophosphate, a non-sterol intermediate of the mevalonate pathway, serves as the substrate for protein geranylgeranylation, a process catalyzed by geranylgeranyl transferase I (GGTase-I). Myeloid-specific deletion of *Pggt1b*, the gene coding for GGTase-I, leads to spontaneous and severe erosive arthritis in mice; however, the underlying mechanisms remained unclear. In this study, we demonstrate that arthritis in mice with myeloid-specific *Pggt1b* deficiency is driven by unprenylated GTP-bound small RHO family GTPases, which in turn trigger Pyrin (*Mefv*) inflammasome activation, GSDMD-dependent macrophage pyroptosis, and IL-1β secretion. We show that although *Pggt1b* deficiency leads to hyperactivation of RAC1, impaired prenylation alters its proper membrane localization and interaction with effectors, rendering it effectively inactive in vivo. Consequently, unprenylated RHO family signaling promotes Pyrin inflammasome assembly through recruitment to the RAC1 effector IQGAP1. Together, these findings identify a novel inflammatory axis in which non-prenylated RHO GTPase activity promotes spontaneous Pyrin inflammasome activation, pyroptosis, and IL-1β release in macrophages, contributing to inflammatory arthritis in mice.**

**Keywords** RHO Family GTPases; Prenylation; Pyrin Inflammasome; Pyroptosis; Arthritis
**Subject Categories** Immunology; Metabolism

## Introduction

The mevalonate pathway is a core metabolic pathway that regulates various cellular functions by synthesizing cholesterol and producing crucial isoprenyl lipid intermediates, such as farnesyl pyrophosphate (FPP) and geranylgeranyl pyrophosphate (GGPP) (Goldstein and Brown, 1990). The lipid intermediates FPP and GGPP act as essential substrates for protein lipidation, a critical post-translational modification of hundreds of proteins, including small GTPases of the RAS, RHO, and RAB families (Zhang and Casey, 1996). Lipidation of many small GTPases typically occurs at a cysteine residue near the C-terminus. RHO and RAS family proteins contain a carboxyterminal *-CAAX* motif (where "C" is cysteine, "A" represents an aliphatic amino acid, and "X" can be any amino acid). Farnesyltransferase-I (FTase-I or *Fntb*) and geranylgeranyltransferase-I (GGTase-I or *Pggt1b*) recognize these *-CAAX* sequences at the C-terminus and subsequently catalyze the transfer of the respective lipid moiety present in FPP and GGPP to the C-terminal cysteine, a process called "prenylation." Prenylation enhances the hydrophobicity and supports membrane anchoring of RHO and RAS proteins, thereby promoting crucial protein–protein and protein–membrane interactions. Hence, the small GTPases' activity, stability, and turnover are profoundly influenced and regulated (Chinpaisal et al, 1997; Hori et al, 1991; Pereira-Leal et al, 2001; Sahai and Marshall, 2002; Solski et al, 2002).

RHO family proteins regulate a wide range of cellular processes, including inflammatory responses (Heasman and Ridley, 2008). While inhibiting the prenylation of RHO family proteins was once considered a promising strategy to prevent the development of inflammatory diseases (Connor et al, 2006; Nagashima et al, 2006; Walters et al, 2002), the conditional deletion of *Pggt1b* in myeloid cells—which abolishes protein prenylation—was unexpectedly found to cause spontaneous and severe erosive arthritis in mice (Khan et al, 2011). *Pggt1b* deficiency in macrophages leads to accumulation of GTP-bound RHO family proteins, resulting in heightened inflammatory responses and increased interleukin (IL)-1β secretion (Akula et al, 2016; Khan et al, 2011).

In this study, we demonstrate that inflammation and erosive arthritis in mice with myeloid-specific *Pggt1b* deficiency are driven by unprenylated RHO GTPase signaling. This leads to impaired membrane localization of hyperactive RHO GTPases, rendering them effectively inactive, and promotes activation of the Pyrin inflammasome, GSDMD-dependent macrophage pyroptosis, and IL-1β secretion in vivo. Genetic deletion of *Il1r1*, *Pycard* (encoding ASC), *Casp1*, *Gsdmd*, or *Mefv* (the gene encoding Pyrin), but not *Nlrp3*, rescued the arthritic phenotype caused by myeloid-restricted *Pggt1b* deficiency.

[1]VIB Center for Inflammation Research, VIB, Ghent, Belgium. [2]Department of Biomedical Molecular Biology, Ghent University, Ghent, Belgium. [3]Department of Rheumatology, Ghent University Hospital, Ghent, Belgium. [4]Department of Internal Medicine and Pediatrics, Ghent University, Ghent, Belgium. [5]College of Health and Life Sciences, Hamad Bin Khalifa University, Doha, Qatar. [6]Department of Medicine, Huddinge, Karolinska Institutet, Huddinge, Sweden. [7]Cell Death Signaling Laboratory, Department of Biomedical Sciences, University of Antwerp, B-Antwerp, Belgium. ✉E-mail: Naga.Akula@irc.vib-ugent.be; geert.vanloo@irc.vib-ugent.be

We show that *Pggt1b* deficiency causes unprenylated RAC1 signaling, which promotes Pyrin inflammasome activation through the recruitment of Pyrin to the RAC1 effector IQGAP1.

## Results

### IL-1β, but not tumor necrosis factor (TNF), drives arthritis development in *Pggt1b*[Δ/Δ] mice

Mice with myeloid-specific *Pggt1b* deficiency (*Pggt1b*[Δ/Δ] mice, also referred to as GGTase-I knockout mice) spontaneously develop severe erosive arthritis (Khan et al, 2011). Their macrophages exhibit increased NF-κB and MAPK activation inducing enhanced expression of pro-inflammatory mediators, including IL-1β, TNF, and IL-6 (Khan et al, 2011). These cytokines have all been implicated in the pathophysiology of rheumatoid arthritis (McInnes et al, 2016), and especially anti-TNF biological agents are the anti-cytokine therapy of choice for the treatment of the disease (Kalliolias and Ivashkiv, 2016; van Loo and Bertrand, 2023). To investigate the importance of TNF in the development of arthritis in *Pggt1b*[Δ/Δ] mice, we genetically deleted *Tnfr1* in *Pggt1b*[Δ/Δ] mice. Histological analyses of knee joints of 20-week-old *Pggt1b*[Δ/Δ]*Tnfr1*[−/−] mice still revealed inflammation, characterized by infiltration of leukocytes in the synovium (synovitis) and accumulation of leukocytes in the joint cavities (Fig. EV1A,B), demonstrating that TNF-R1 signaling is not driving the inflammatory pathology in *Pggt1b*[Δ/Δ] mice.

Since we previously showed that *Pggt1b* deficiency induces IL-1β release in macrophages (Akula et al, 2019; Akula et al, 2016), we next addressed the importance of IL-1 signaling for driving arthritis pathology in vivo. For this, we crossed *Pggt1b*[Δ/Δ] mice with IL-1-receptor 1 (*Il-1r1*)-deficient mice. While GGTase-I knockout mice (*Pggt1b*[Δ/Δ]*Il-1r1*[+/+]) all developed arthritis, as is evidenced by the presence of inflammatory cells in the knee joint, *Pggt1b*[Δ/Δ]*Il-1r1*[−/−] mice were completely protected from disease development (Fig. 1A,B), demonstrating that IL-1β is the driving cytokine for arthritis pathology in myeloid GGTase-I-deficient mice. The importance of IL-1β signaling for arthritis development was further confirmed by crossing *Pggt1b*[Δ/Δ] mice with mice deficient in the IL-1 receptor adapter protein MyD88. *Pggt1b*[Δ/Δ]*Myd88*[−/−] mice were fully protected from arthritis development (Fig. 1C,D), similar to what was observed in IL-1R1-deficient conditions. Together, these results demonstrate that IL-1R1-MyD88 signaling, activated in response to IL-1β release, drives the inflammatory disease in *Pggt1b*[Δ/Δ] mice.

### Inflammasome-induced GSDMD-mediated pyroptosis drives IL-1β production in *Pggt1b*[Δ/Δ] macrophages

IL-1β maturation usually depends on the activity of inflammasomes. In addition to processing pro-IL-1β, inflammasomes also trigger pyroptosis, a lytic cell death mode that facilitates the release of IL-1β and other inflammatory mediators into the environment. Pyroptosis happens through cleavage of the executioner protein Gasdermin D (GSDMD), after which the cleaved amino-terminal GSDMD domain oligomerizes and perforates the plasma membrane to induce cell swelling and osmotic lysis of macrophages (de Vasconcelos and Lamkanfi, 2020; Shi et al, 2015). We hypothesized that inflammasome-induced GSDMD-mediated pyroptosis plays a

critical role in driving the inflammatory phenotype observed in *Pggt1b*[Δ/Δ] mice. Unlike in wildtype (*Pggt1b*[+/+]) bone marrow-derived macrophages (BMDMs), lipopolysaccharide (LPS) stimulation alone was sufficient to induce robust caspase-1 and pro IL-1β cleavage, and release of mature IL-1β from *Pggt1b*[Δ/Δ] macrophages (Fig. 2A,B). LPS-stimulated *Pggt1b*[Δ/Δ] macrophages also exhibited significantly increased cell death compared to *Pggt1b*[+/+] macrophages (Fig. 2C). We confirmed GSDMD cleavage in *Pggt1b*[Δ/Δ] macrophages in response to LPS, whereas no such cleavage could be observed in LPS-stimulated *Pggt1b*[+/+] macrophages (Fig. 2A). These results suggest that GSDMD-mediated pyroptosis may be responsible for the release of bioactive IL-1β from *Pggt1b*[Δ/Δ] macrophages. To validate the role of inflammasome activation and GSDMD-induced pyroptosis in the release of bioactive IL-1β from *Pggt1b*[Δ/Δ] macrophages, we next deleted *Pycard* (encoding the inflammasome adapter protein ASC), *Casp1*, or *Gsdmd* in *Pggt1b*[Δ/Δ] macrophages. While *Pggt1b*[Δ/Δ] macrophages exhibited caspase-1 activation, proIL-1β cleavage, GSDMD cleavage, and IL-1β release in response to LPS stimulation, genetic ablation of *Pycard*, *Casp1*, or *Gsdmd* effectively suppressed these responses (Fig. 2D–I). To assess the contribution of other regulated cell death pathways, viz., apoptosis and necroptosis, to IL-1β secretion in *Pggt1b*[Δ/Δ] macrophages, we deleted *Casp8* and *Ripk3*, or genetically inactivated RIPK1 kinase activity. However, none of these interventions affected caspase-1 activation, GSDMD cleavage, or IL-1β maturation and extracellular release in *Pggt1b*[Δ/Δ] macrophages (Fig. EV2A–D). Collectively, these results demonstrate that *Pggt1b* deficiency in macrophages selectively elicits inflammasome-induced pyroptosis and IL-1β production and secretion in response to LPS stimulation alone.

### Pyrin mediates inflammasome activation and IL-1β secretion in *Pggt1b*[Δ/Δ] macrophages

Knockdown of Pyrin (*Mefv*) using small interfering RNA (siRNA) has previously been shown to reduce inflammasome activation and IL-1β release in LPS-primed *Pggt1b*[Δ/Δ] macrophages (Akula et al, 2016). Pyrin expression is transcriptionally regulated (Cornut et al, 2020). While LPS stimulation upregulates Pyrin expression in wildtype BMDMs, we observed slightly elevated basal levels and consistently higher Pyrin protein expression following LPS stimulation in *Pggt1b*[Δ/Δ] macrophages compared to wildtype macrophages (Fig. 3A). In contrast, Nlrp3 protein levels were similarly induced by LPS stimulation in both wildtype and *Pggt1b*[Δ/Δ] macrophages (Fig. 3A). Interestingly, genetic deletion of *Myd88* normalized Pyrin protein levels in *Pggt1b*[Δ/Δ] macrophages (Fig. 3B), indicating that GGTase-I deficiency enhances transcriptional upregulation of Pyrin in a MyD88-dependent manner. Moreover, blocking IL-1 receptor (IL-1R) signaling, using a recombinant mouse IL-1R antagonist, significantly diminished LPS-induced Pyrin expression levels in *Pggt1b*[Δ/Δ] macrophages, suggesting that IL-1β secretion from GGTase-I knockout macrophages contributes importantly to elevated Pyrin expression levels as part of a positive feedback loop (Fig. 3C).

To confirm the importance of Pyrin for inflammasome activation and IL-1β production in *Pggt1b*[Δ/Δ] macrophages, we next examined the effect of *Mefv* deletion in *Pggt1b*[Δ/Δ] macrophages. Genetic deletion of *Mefv* blunted LPS-induced caspase-1 activation, GSDMD cleavage, and IL-1β processing in *Pggt1b*[Δ/Δ] macrophages

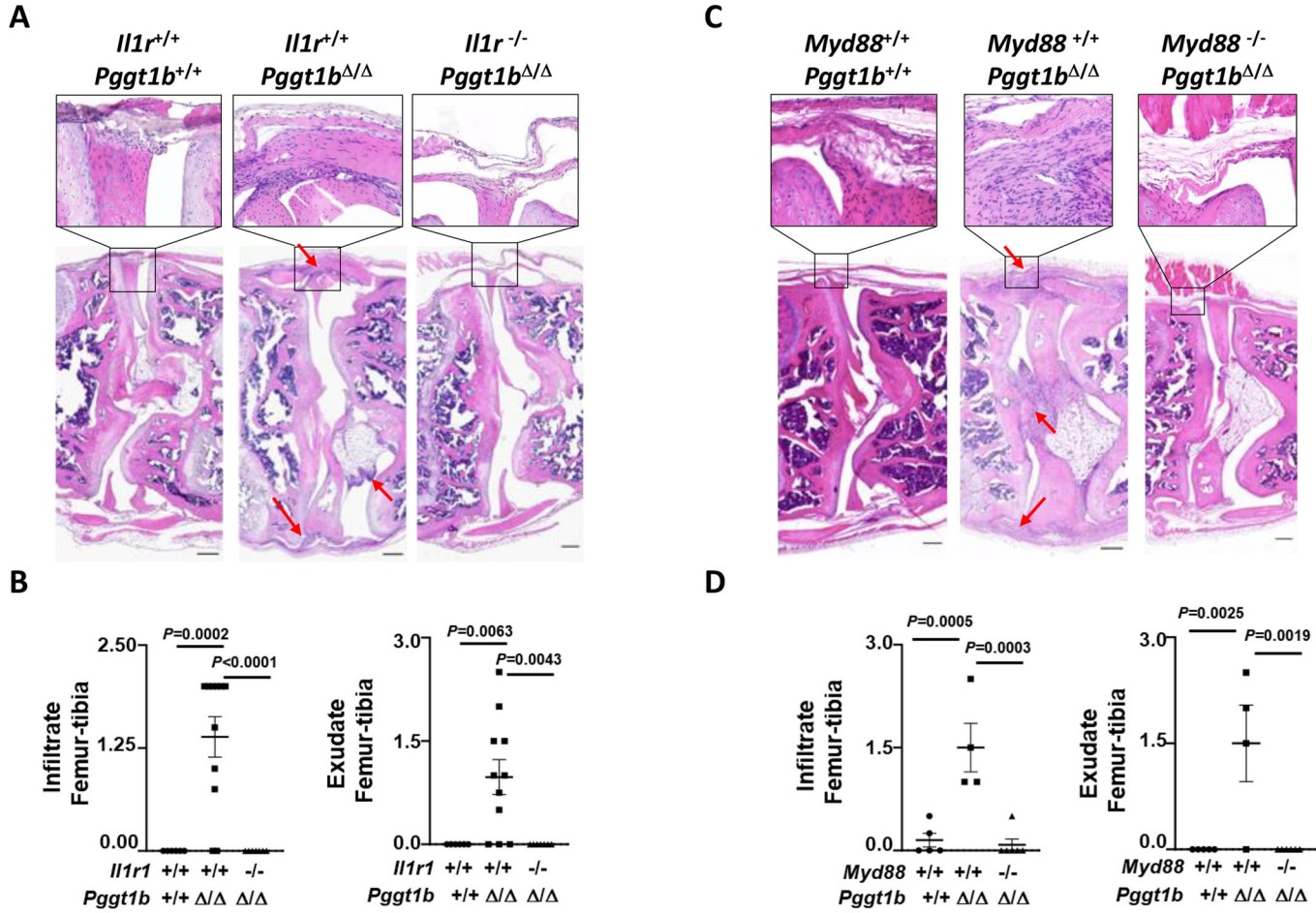

**Figure 1. MyD88-dependent IL-1β signaling mediates arthritis development in Pggt1b$^{\Delta/\Delta}$ mice.**

(A) Histological images of haematoxylin and eosin-stained knee joints of 20-week-old *Il1r1$^{+/+}$Pggt1b$^{+/+}$*, *Il1r1$^{+/+}$Pggt1b$^{\Delta/\Delta}$*, and *Il1r1$^{-/-}$Pggt1b$^{\Delta/\Delta}$* mice. Representative pictures are shown. Scalebar, 200 μm. Arrows depict inflammation as shown by the accumulation of infiltrating leukocytes. (B) Infiltrate and exudate femur and tibia scores, each ranging from 0 (normal) to 3 (severely inflamed), of *Il1r1$^{+/+}$Pggt1b$^{+/+}$* (*n* = 6), *Il1r1$^{+/+}$Pggt1b$^{\Delta/\Delta}$* (*n* = 11), *Il1r1$^{-/-}$Pggt1b$^{\Delta/\Delta}$* (*n* = 7). Dots in the graphs indicate individual mice, and data are expressed as mean ± s.e.m. (C) Histological images of haematoxylin and eosin-stained knee joints of 20-week-old *Myd88$^{+/+}$Pggt1b$^{+/+}$*, *Myd88$^{+/+}$Pggt1b$^{\Delta/\Delta}$*, and *Myd88$^{-/-}$Pggt1b$^{\Delta/\Delta}$* mice. Representative pictures are shown. Scalebar, 200 μm. Arrows depict inflammation. (D) Infiltrate and exudate femur and tibia scores, each ranging from 0 (normal) to 3 (severely inflamed), of *Myd88$^{+/+}$Pggt1b$^{+/+}$* (*n* = 5), *Myd88$^{+/+}$Pggt1b$^{\Delta/\Delta}$* (*n* = 4), and *Myd88$^{-/-}$Pggt1b$^{\Delta/\Delta}$* (*n* = 6). Dots in the graphs indicate individual mice, and data are expressed as mean ± s.e.m. Significance between groups was calculated by one-way ANOVA and Dunnett's multiple comparison test. Source data are available online for this figure.

(Fig. 3D,E). In marked contrast, ablation of *Nlrp3* did not prevent inflammasome activation or IL-1β release in *Pggt1b$^{\Delta/\Delta}$* macrophages (Fig. 3F,G). Together, these results confirm that the Pyrin inflammasome, but not the Nlrp3 inflammasome, mediates LPS-induced IL-1β production in GGTase-I-deficient macrophages.

## Pyrin-mediated GSDMD signaling drives arthritis development in Pggt1b$^{\Delta/\Delta}$ mice

To demonstrate the in vivo importance of inflammasome activation and pyroptosis for the development of the erosive arthritis phenotype, we bred myeloid-specific *Pggt1b* knockout mice with mice deficient for ASC (*Pggt1b$^{\Delta/\Delta}$Asc$^{-/-}$*), caspase-1 (*Pggt1b$^{\Delta/\Delta}$Casp1$^{-/-}$*), or GSDMD (*Pggt1b$^{\Delta/\Delta}$Gsdmd$^{-/-}$*). In contrast to *Pggt1b$^{\Delta/\Delta}$* mice, which all developed arthritis pathology, *Pggt1b$^{\Delta/\Delta}$ Asc$^{-/-}$*, *Pggt1b$^{\Delta/\Delta}$Casp1$^{-/-}$*, and *Pggt1b$^{\Delta/\Delta}$Gsdmd$^{-/-}$* mice were

protected from disease development, as evidenced by the absence of inflammatory cells in the knee joints of these compound knockout mice (Fig. 4A–F). IL-1β is notoriously challenging to measure in blood samples; however, serum levels of the inflammasome-dependent cytokine IL-18 closely mirrored our histology-based assessment of arthritis and were markedly blunted in inflammasome-deficient mice (Fig. EV3). These results demonstrate that inflammasome activation and GSDMD-induced pyroptosis are crucial for the development of the erosive arthritis phenotype of *Pggt1b$^{\Delta/\Delta}$* mice. Consistent with our ex vivo findings in BMDMs (Fig. EV2), inactivation of RIPK1 kinase activity, achieved by crossing *Pggt1b$^{\Delta/\Delta}$* mice with *Ripk1$^{D138N/D138N}$* mice, did not prevent arthritis development in *Pggt1b$^{\Delta/\Delta}$* mice (Fig. EV4A–C).

Next, to determine whether Pyrin serves as the inflammasome sensor driving arthritis in *Pggt1b$^{\Delta/\Delta}$* mice, we crossed the latter onto a *Mefv*-deficient background. In contrast to GGTase-I-deficient

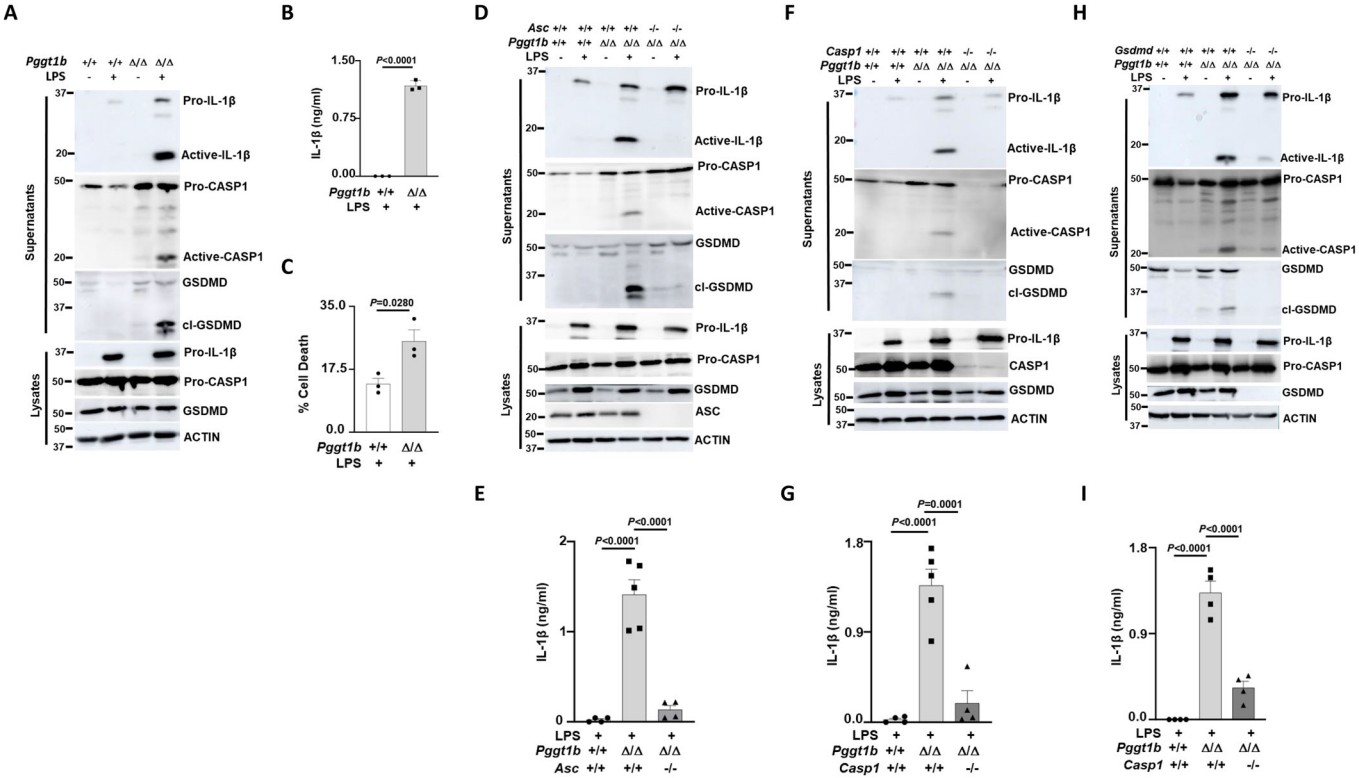

**Figure 2. GSDMD-mediated pyroptosis drives IL-1β production in Pggt1b^{Δ/Δ} macrophages.**

(A) Western blots showing levels of pro- and bioactive IL-1β, pro- and active caspase-1 (CASP1), and full length and cleaved (cl) GSDMD in supernatants and lysates of LPS-stimulated bone marrow-derived macrophages (BMDMs) isolated from Pggt1b^{+/+} and Pggt1b^{Δ/Δ} mice. Actin was used as a loading control. (B) IL-1β cytokine levels, 8 h after LPS stimulation, in supernatants of BMDMs isolated from Pggt1b^{+/+} (n = 3 biological replicates) and Pggt1b^{Δ/Δ} (n = 3) mice. (C) Percentage (%) of cell death measured as LDH release in supernatants 8 h after LPS stimulation of BMDMs isolated from Pggt1b^{+/+} (n = 3 biological replicates) and Pggt1b^{Δ/Δ} (n = 3) mice. (D) Western blots showing levels of pro- and bioactive IL-1β, pro- and active CASP1, and full length and cleaved (cl) GSDMD in supernatants and lysates of LPS-stimulated BMDMs isolated from Pggt1b^{+/+}, Pggt1b^{Δ/Δ}, and Asc^{−/−}Pggt1b^{Δ/Δ} mice. Actin was used as a loading control. (E) IL-1β cytokine levels, 8 h after LPS stimulation, in supernatants of BMDMs isolated from Pggt1b^{+/+} (n = 4 biological replicates), Pggt1b^{Δ/Δ} (n = 5), and Asc^{−/−}Pggt1b^{Δ/Δ} (n = 4) mice. (F) Western blots showing levels of pro- and bioactive IL-1β, pro- and active CASP1, and full length and cleaved (cl) GSDMD in supernatants and lysates of LPS-stimulated BMDMs isolated from Pggt1b^{+/+}, Pggt1b^{Δ/Δ}, and Casp1^{−/−}Pggt1b^{Δ/Δ} mice. Actin was used as a loading control. (G) IL-1β cytokine levels, 8 h after LPS stimulation, in supernatants of BMDMs isolated from Pggt1b^{+/+} (n = 4 biological replicates), Pggt1b^{Δ/Δ} (n = 5), and Casp1^{−/−}Pggt1b^{Δ/Δ} (n = 4) mice. (H) Western blots showing levels of pro- and bioactive IL-1β, pro- and active CASP1, and full length and cleaved (cl) GSDMD in supernatants and lysates of LPS-stimulated BMDMs isolated from Pggt1b^{+/+}, Pggt1b^{Δ/Δ}, and Gsdmd^{−/−}Pggt1b^{Δ/Δ} mice. Actin was used as a loading control. (I) IL-1β cytokine levels, 8 h after LPS stimulation, in supernatants of BMDMs isolated from Pggt1b^{+/+} (n = 4 biological replicates), Pggt1b^{Δ/Δ} (n = 4), and Gsdmd^{−/−}Pggt1b^{Δ/Δ} (n = 4) mice. Significance between groups was calculated by two-tailed Student's t test (B, C), one-way ANOVA with Dunnett's multiple comparisons test (E, G, I). Data are expressed as mean ± s.e.m. Western blots are representative of two independent experiments using biological replicates. Source data are available online for this figure.

mice (Pggt1b^{Δ/Δ}Mefv^{+/+}), Pyrin-deficient Pggt1b^{Δ/Δ} mice (Pggt1b^{Δ/Δ}Mefv^{−/−}) were protected from developing arthritis, consistent with their reduced serum IL-18 levels (Figs. 4G,H and EV3). In contrast, genetic ablation of the inflammasome sensor Nlrp3 in Pggt1b^{Δ/Δ} mice failed to prevent arthritis development (Fig. EV4D–F). These results clearly demonstrate that Pyrin, rather than Nlrp3, is the key inflammasome sensor driving arthritis pathogenesis in Pggt1b^{Δ/Δ} mice.

## RAC1 activates the Pyrin inflammasome in Pggt1b^{Δ/Δ} macrophages

In response to RHOA GTPase inactivation, such as that induced by Clostridium difficile TcdA or TcdB treatment, Pyrin inflammasome activation follows a two-step activation process. First, Pyrin is dephosphorylated and dissociates from the chaperone protein 14-3-3.

In the second step, conformational changes expose the PYD domain of Pyrin, allowing it to interact with the adapter protein ASC and initiate assembly of the Pyrin inflammasome (Gao et al, 2016; Park et al, 2016; Xu et al, 2014). Additionally, previous work has demonstrated that GGTase-I deficiency in macrophages leads to the hyperactivation of RHO family proteins, particularly RAC1-GTP, RHOA-GTP, and CDC42-GTP (Akula et al, 2019; Khan et al, 2011). To address the importance of RHO family signaling and the nature of the RHO family member for Pyrin inflammasome activation in GGTase-I-deficient macrophages, we treated Pggt1b^{Δ/Δ} macrophages with either the RAC1 inhibitor EHT1864 (Onesto et al, 2008; Shutes et al, 2007) or with the RHOA inhibitor CCG1423 (Evelyn et al, 2007). Pharmacological inhibition of RAC1 with EHT1864 abrogated LPS-induced IL-1β secretion (Fig. 5A) and also suppressed pro-IL-1β processing, pro-caspase-1 cleavage, and GSDMD cleavage in Pggt1b^{Δ/Δ} macrophages, in a dose-dependent manner (Fig. 5B). In contrast, inhibition of

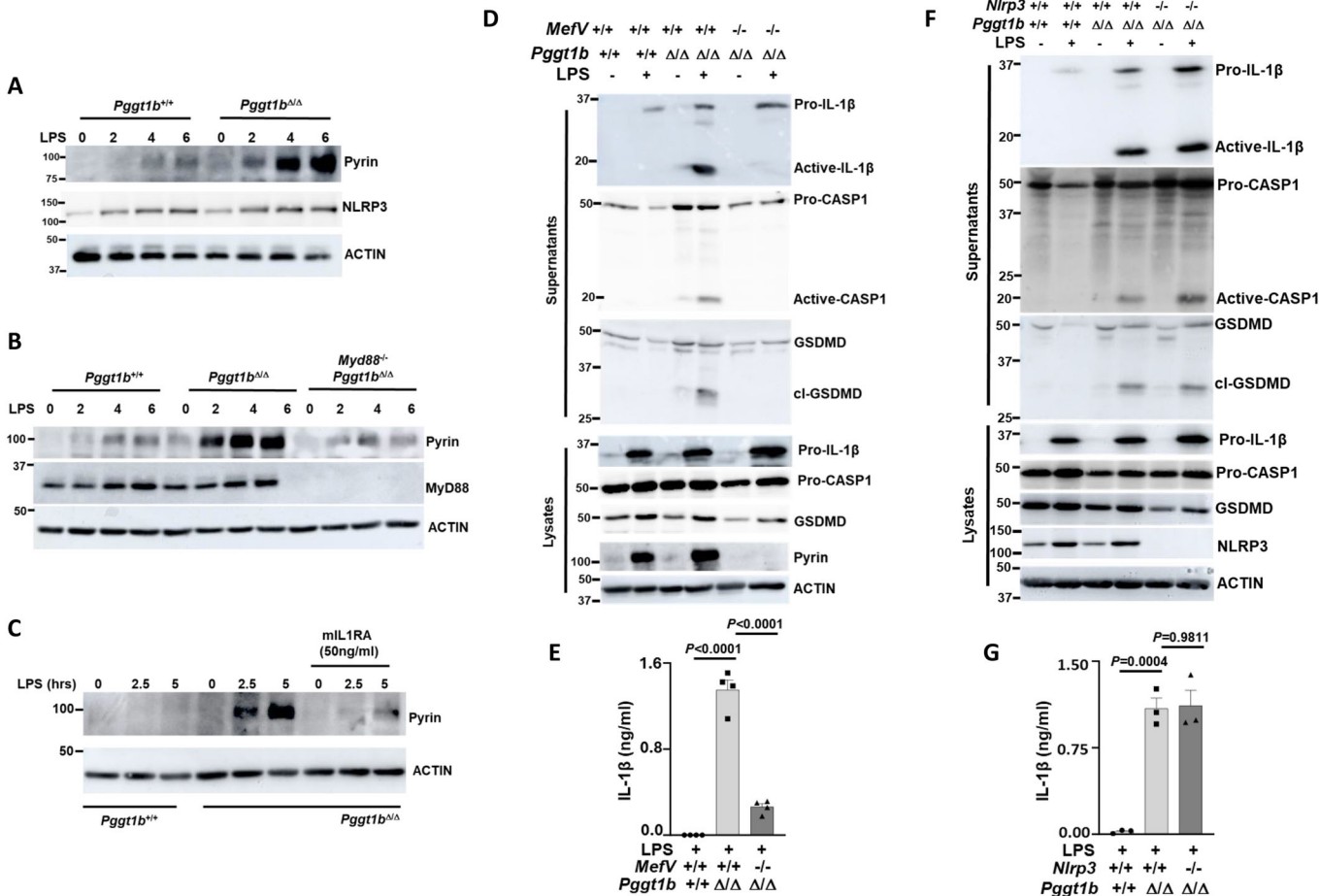

**Figure 3. Pyrin but not Nlrp3 mediates inflammasome activation in $Pggt1b^{\Delta/\Delta}$ macrophages.**

(A) Western blots showing levels of Pyrin and Nlrp3 in lysates from LPS-stimulated BMDMs isolated from $Pggt1b^{+/+}$ and $Pggt1b^{\Delta/\Delta}$ mice. Actin was used as a loading control. (B) Western blots showing levels of Pyrin in lysates from LPS-stimulated BMDMs isolated from $Pggt1b^{+/+}$, $Pggt1b^{\Delta/\Delta}$, and $Myd88^{-/-}Pggt1b^{\Delta/\Delta}$ mice. Actin was used as a loading control. (C) Western blots showing levels of Pyrin in lysates from LPS-stimulated $Pggt1b^{+/+}$ and $Pggt1b^{\Delta/\Delta}$ macrophages, in the presence or absence of recombinant mouse interleukin-1 receptor (IL-1R) antagonist (50 ng/ml). Actin was used as a loading control. (D) Western blots showing levels of pro- and bioactive-IL-1β, pro- and active-CASP1 and full length and cleaved (cl) GSDMD in supernatants and lysates from LPS-stimulated BMDMs isolated from $Pggt1b^{+/+}$, $Pggt1b^{\Delta/\Delta}$ and $Mefv^{-/-}Pggt1b^{\Delta/\Delta}$ macrophages. Actin was used as a loading control. (E) IL-1β cytokine levels, 8 h after LPS stimulation, in supernatants isolated from $Pggt1b^{+/+}$ ($n = 4$ biological replicates), $Pggt1b^{\Delta/\Delta}$ ($n = 4$), and $Mefv^{-/-}Pggt1b^{\Delta/\Delta}$ ($n = 4$) macrophages. Data are expressed as mean ± s.e.m. (F) Western blots showing levels of pro- and bioactive-IL-1β, pro- and active-CASP1 and full length and cleaved (cl) GSDMD in supernatants and lysates from LPS-stimulated BMDMs isolated from $Pggt1b^{+/+}$, $Pggt1b^{\Delta/\Delta}$, and $Nlrp3^{-/-}Pggt1b^{\Delta/\Delta}$ macrophages. Actin was used as a loading control. (G) IL-1β cytokine levels, 8 h after LPS stimulation, in supernatants isolated from $Pggt1b^{+/+}$ ($n = 3$ biological replicates), $Pggt1b^{\Delta/\Delta}$ ($n = 3$), and $Nlrp3^{-/-}Pggt1b^{\Delta/\Delta}$ ($n = 3$) macrophages. Data are expressed as mean ± s.e.m. Significance between groups was calculated by one-way ANOVA and Dunnett's multiple comparison test. Western blots are representative of two independent experiments (except for C) using biological replicates. Source data are available online for this figure.

RHOA with CCG1423 had no effect on these responses (Fig. 5A,C). RAC1 inhibition also normalized Pyrin protein levels in LPS-treated $Pggt1b^{\Delta/\Delta}$ macrophages (Fig. 5B). As a control, we confirmed that EHT1864 treatment reduces RAC1-GTP, but not RHOA-GTP, levels in $Pggt1b^{\Delta/\Delta}$ macrophages, indicating that the suppressive effects seen with EHT1864 treatment are mediated by RAC1 inhibition (Fig. EV5A,B). To assess whether GGTase deficiency directly affects RHOA activity, we pulled down RHOA-GTP from LPS-treated $Pggt1b$ knockout and wildtype BMDMs, confirming RHOA hyperactivation (GTP-bound RHOA) in $Pggt1b$ knockout macrophages upon LPS stimulation (Fig. EV5C). Finally, in addition to EHT1864, pharmacological inhibition of the RAC1 effector p38 MAP kinase using

SB203580 also abrogated LPS-induced IL-1β secretion, as well as pro-IL-1β processing, and caspase-1 and GSDMD cleavage (Fig. 5D,E). Together, these results demonstrate that unprenylated RAC1 signaling plays a crucial role in promoting Pyrin inflammasome activation in $Pggt1b^{\Delta/\Delta}$ macrophages.

## Microtubule dynamics mediate Pyrin inflammasome activation in $Pggt1b^{\Delta/\Delta}$ macrophages

Association of Pyrin with microtubules is a critical step in its activation, as it promotes conformational changes that enable interaction with the adapter protein ASC. Accordingly, colchicine,

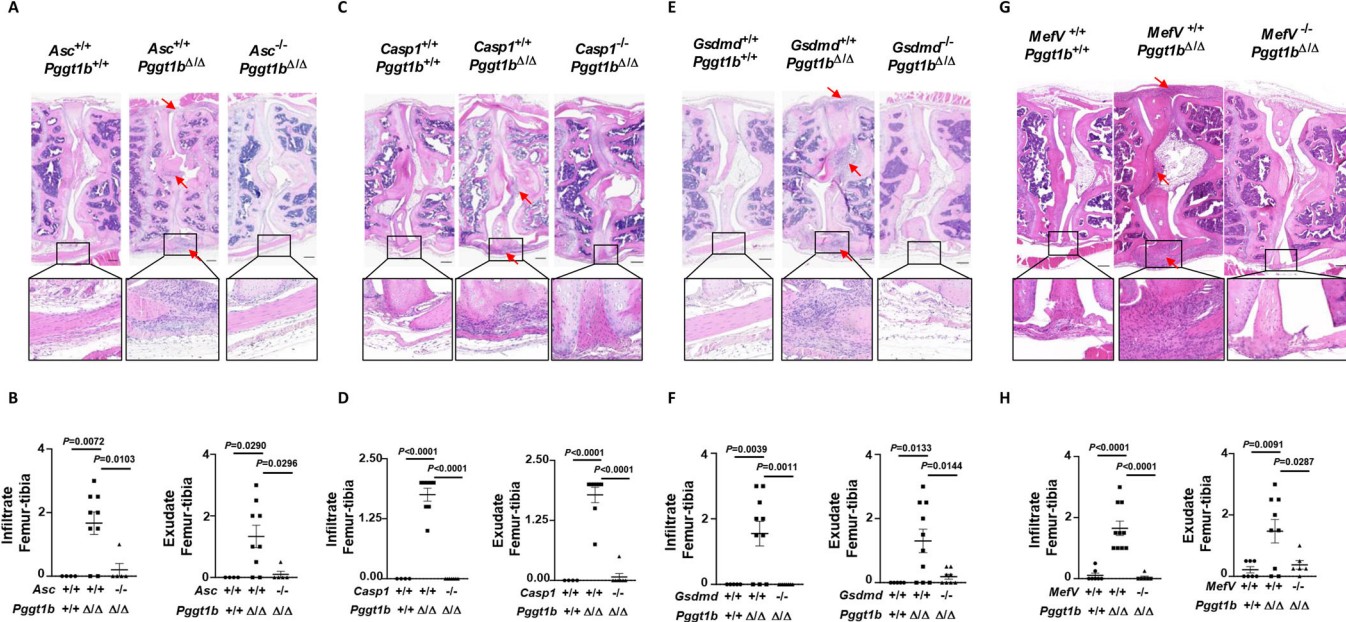

**Figure 4. Pyrin-mediated GSDMD signaling mediates arthritis development in *Pggt1b*$^{\Delta/\Delta}$ mice.**

(A) Histological images of haematoxylin and eosin-stained knee joints of 20-week-old *Pggt1b*$^{+/+}$, *Pggt1b*$^{\Delta/\Delta}$, and *Asc*$^{-/-}$*Pggt1b*$^{\Delta/\Delta}$ mice. Representative pictures are shown. Scalebar, 200 μm. Arrows depict inflammation, as shown by the accumulation of infiltrating leukocytes. (B) Infiltrate and exudate femur and tibia scores, each ranging from 0 (normal) to 3 (severely inflamed), of *Asc*$^{+/+}$*Pggt1b*$^{+/+}$ ($n = 4$), *Asc*$^{+/+}$*Pggt1b*$^{\Delta/\Delta}$ ($n = 9$), and *Asc*$^{-/-}$*Pggt1b*$^{\Delta/\Delta}$ ($n = 5$) at the age of 20 weeks. Dots in the graphs indicate individual mice, and data are expressed as mean ± s.e.m. (C) Histological images of haematoxylin and eosin-stained knee joints of *Pggt1b*$^{+/+}$, *Pggt1b*$^{\Delta/\Delta}$, and *Casp1*$^{-/-}$*Pggt1b*$^{\Delta/\Delta}$ mice. Representative pictures are shown. Scalebar, 200 μm. Arrows depict inflammation. (D) Infiltrate and exudate femur and tibia scores, each ranging from 0 (normal) to 3 (severely inflamed), of *Casp1*$^{+/+}$*Pggt1b*$^{+/+}$ ($n = 4$), *Casp1*$^{+/+}$*Pggt1b*$^{\Delta/\Delta}$ ($n = 8$), and *Casp1*$^{-/-}$*Pggt1b*$^{\Delta/\Delta}$ ($n = 7$) at the age of 20 weeks. Dots in the graphs indicate individual mice, and data are expressed as mean ± s.e.m. (E) Histological images of haematoxylin and eosin-stained knee joints of *Pggt1b*$^{+/+}$, *Pggt1b*$^{\Delta/\Delta}$, and *Gsdmd*$^{-/-}$*Pggt1b*$^{\Delta/\Delta}$ mice. Representative pictures are shown. Scalebar, 200 μm. Arrows depict inflammation. (F) Infiltrate and exudate femur and tibia scores, each ranging from 0 (normal) to 3 (severely inflamed), of *Gsdmd*$^{+/+}$*Pggt1b*$^{+/+}$ ($n = 5$), *Gsdmd*$^{+/+}$*Pggt1b*$^{\Delta/\Delta}$ ($n = 10$), and *Gsdmd*$^{-/-}$*Pggt1b*$^{\Delta/\Delta}$ ($n = 8$) at the age of 20 weeks. Dots in the graphs indicate individual mice, and data are expressed as mean ± s.e.m. (G) Histological images of haematoxylin and eosin-stained knee joints of *Pggt1b*$^{+/+}$, *Pggt1b*$^{\Delta/\Delta}$, and *Mefv*$^{-/-}$*Pggt1b*$^{\Delta/\Delta}$ mice. Representative pictures are shown. Scalebar, 200 μm. Arrows depict inflammation. (H) Infiltrate and exudate femur and tibia scores, each ranging from 0 (normal) to 3 (severely inflamed), of *Mefv*$^{+/+}$*Pggt1b*$^{+/+}$ ($n = 7$), *Mefv*$^{+/+}$*Pggt1b*$^{\Delta/\Delta}$ ($n = 9$), and *Mefv*$^{-/-}$*Pggt1b*$^{\Delta/\Delta}$ ($n = 7$) at the age of 20 weeks. Dots in the graphs indicate individual mice, and data are expressed as mean ± s.e.m. Significance between groups was calculated by one-way ANOVA and Dunnett's multiple comparison test. Source data are available online for this figure.

a microtubule polymerization inhibitor, blocks Pyrin inflammasome activation downstream of Pyrin dephosphorylation (Gao et al, 2016; Van Gorp et al, 2016). To assess the role of microtubule dynamics in Pyrin inflammasome activation in *Pggt1b*$^{\Delta/\Delta}$ macrophages, we treated wildtype and *Pggt1b*$^{\Delta/\Delta}$ macrophages with colchicine. In marked contrast to untreated cells, colchicine pretreatment abolished caspase-1 maturation, GSDMD cleavage, and IL-1β secretion in LPS-stimulated *Pggt1b*$^{\Delta/\Delta}$ BMDMs (Fig. 6A,B), confirming that intact microtubules are essential for Pyrin inflammasome activation in *Pggt1b*$^{\Delta/\Delta}$ macrophages.

Pyrin phosphorylation by the protein kinase C (PKC) family kinases PKN1 and PKN2 inhibits Pyrin activation in macrophages (Gao et al, 2016; Magnotti et al, 2019; Masters et al, 2016; Park et al, 2016; Van Gorp et al, 2016). In line with the increased LPS-induced Pyrin expression observed in *Pggt1b*$^{\Delta/\Delta}$ macrophages, we also detected elevated levels of phosphorylated Pyrin (Fig. EV6A). This suggests that, despite the rise in total Pyrin protein levels, the ratio of phosphorylated to total Pyrin remains relatively similar between control and *Pggt1b*$^{\Delta/\Delta}$ macrophages. In contrast, treatment with the Clostridial toxin TcdB triggers Pyrin dephosphorylation, not only

in wildtype macrophages, as previously shown (Park et al, 2016; Xu et al, 2014), but also in *Pggt1b*$^{\Delta/\Delta}$ macrophages (Fig. EV6B). These results suggest that GGTase-I deficiency does not modulate Pyrin dephosphorylation directly.

Based on our observation that LPS stimulation suffices to induce elevated Pyrin expression and induce inflammasome activation in *Pggt1b*$^{\Delta/\Delta}$ macrophages, we hypothesized that GGTase-I deficiency may enhance Pyrin inflammasome activation upon suppressing PKC kinase-mediated Pyrin phosphorylation. To test this hypothesis, we induced PKC activation by stimulating *Pggt1b*$^{\Delta/\Delta}$ macrophages with the PKC activators arachidonic acid (AA) and bryostatin (BST). Consistent with our hypothesis, treatment with AA and BST prevented LPS-induced pro-IL-1β processing and secretion, as well as caspase-1 cleavage and GSDMD cleavage, in *Pggt1b*$^{\Delta/\Delta}$ macrophages (Fig. 6C–F). In addition, PKC activation normalized Pyrin expression levels in *Pggt1b*$^{\Delta/\Delta}$ macrophages (Fig. 6C,E), consistent with our earlier results showing that IL-1β secretion contributes importantly to elevated Pyrin expression levels as part of a positive feedback loop (Fig. 3C).

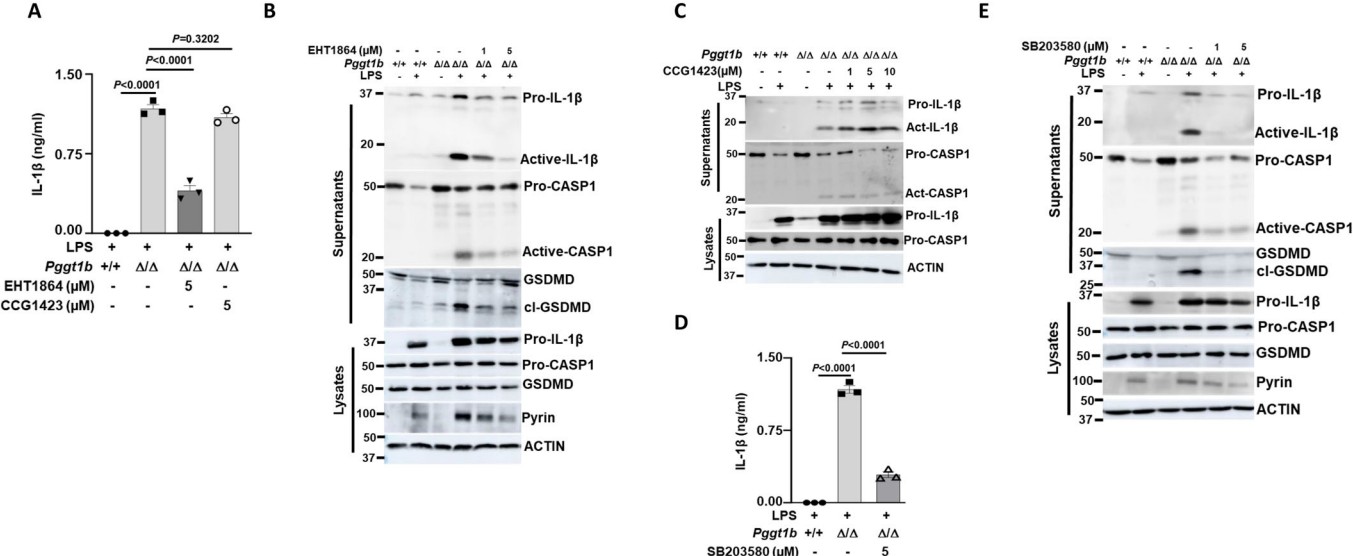

**Figure 5. RAC1 but not RHOA activates the Pyrin inflammasome in $Pggt1b^{\Delta/\Delta}$ macrophages.**

(A) IL-1β cytokine levels, 8 h after LPS stimulation, in supernatants of BMDMs isolated from $Pggt1b^{+/+}$ ($n = 3$ biological replicates), and $Pggt1b^{\Delta/\Delta}$ ($n = 3$) mice, after treatment with EHT1864 or CCG1423. (B) Western blots showing levels of pro- and bioactive IL-1β, pro- and active CASP1, and full length and cleaved (cl) GSDMD, in supernatants and lysates of LPS-stimulated BMDMs after treatment with EHT1864. Actin was used as a loading control. (C) Western blots showing levels of pro- and bioactive IL-1β, pro- and active CASP1, and full length and cleaved (cl) GSDMD, in supernatants and lysates of LPS-stimulated BMDMs after treatment with CCG1423. Actin was used as a loading control. (D) IL-1β cytokine levels, 8 h after LPS stimulation, in supernatants of BMDMs, isolated from $Pggt1b^{+/+}$ ($n = 3$ biological replicates) and $Pggt1b^{\Delta/\Delta}$ ($n = 3$) mice, after treatment with SB203580. Data are expressed as mean ± s.e.m. Significance between groups was calculated by one-way ANOVA and Dunnett's multiple comparison test. (E) Western blots showing levels of pro- and bioactive IL-1β, pro- and active CASP1, and full length and cleaved (cl) GSDMD, in supernatants and lysates of LPS-stimulated BMDMs after treatment with SB203580. Actin was used as a loading control. Western blots are representative of two independent experiments (except for C) using biological replicates. Source data are available online for this figure.

## The RAC1 effector IQGAP1 regulates Pyrin inflammasome activation by interacting with Pyrin in $Pggt1b^{\Delta/\Delta}$ macrophages

IQ motif-containing GTPase-activating protein 1 (IQGAP1) is a crucial scaffolding protein that interacts with numerous binding partners and thereby regulates the formation of complexes involved in intracellular signaling and cytoskeletal dynamics (Briggs and Sacks, 2003; Brown and Sacks, 2006; White et al, 2012). RAC1 is among the best characterized IQGAP1-binding partners (Smith et al, 2015), and IQGAP1 inactivation was previously shown to normalize RAC1 GTP-loading (Akula et al, 2019). To investigate the mechanistic role of IQGAP1 in Pyrin inflammasome activation, we employed CRISPR-Cas9-mediated gene editing to delete IQGAP1 in Pggt1b knockout macrophages. Successful deletion of IQGAP1 was confirmed by Western blot analysis (Fig. 7A). Loss of IQGAP1 abrogated LPS-induced IL-1β processing and secretion, as well as caspase-1 activation and GSDMD cleavage, in $Pggt1b^{\Delta/\Delta}$ macrophages (Fig. 7B,C). Additionally, pyroptotic cell death was normalized in $Pggt1b^{\Delta/\Delta}$ macrophages lacking IQGAP1 expression (Fig. 7D). Collectively, these results demonstrate that IQGAP1 plays a critical role in promoting Pyrin inflammasome activation in $Pggt1b^{\Delta/\Delta}$ macrophages. Surprisingly, however, IQGAP1 deletion did not normalize LPS-induced Pyrin expression levels in $Pggt1b^{\Delta/\Delta}$ macrophages (Fig. 7C), suggesting that IQGAP1 specifically regulates Pyrin inflammasome activation in $Pggt1b^{\Delta/\Delta}$ macrophages without modulating its transcriptional upregulation. Based on these observations and previous findings that nonprenylated RAC1

interacts with IQGAP1 (Akula et al, 2019; Raulien et al, 2024), we hypothesized that IQGAP1 may facilitate recruitment of Pyrin to nonprenylated RAC1 in $Pggt1b^{\Delta/\Delta}$ macrophages. In agreement with this hypothesis, co-immunoprecipitation experiments confirmed an interaction between Pyrin and IQGAP1 in $Pggt1b^{\Delta/\Delta}$ macrophages following LPS stimulation (Fig. 7E). Furthermore, treatment with either the RAC1 inhibitor EHT1864 or the protein kinase C activators AA or BST disrupted the association between Pyrin and IQGAP1 in $Pggt1b^{\Delta/\Delta}$ macrophages in response to LPS (Fig. 7F). Together, these findings suggest that IQGAP1 recruits nonprenylated RAC1 to Pyrin to promote Pyrin inflammasome activation, leading to downstream maturation and secretion of IL-1β in LPS-stimulated $Pggt1b^{\Delta/\Delta}$ macrophages.

## Discussion

In this study, we describe and characterize the first spontaneous mouse model of erosive arthritis driven by activation of the Pyrin inflammasome. We identify four key components underlying the pathogenic inflammatory mechanism in mice lacking GGTase-I specifically in macrophages (Fig. EV7). Firstly, we demonstrate that IL-1, rather than TNF signaling, plays a central role in driving arthritis pathogenesis in GGTase-I-deficient mice. Secondly, we identify the Pyrin inflammasome as the key upstream mechanism promoting IL-1-mediated inflammatory pathology in this arthritis model. Thirdly, we reveal the critical contribution of GSDMD-mediated pyroptosis in amplifying the arthritic phenotype. Lastly,

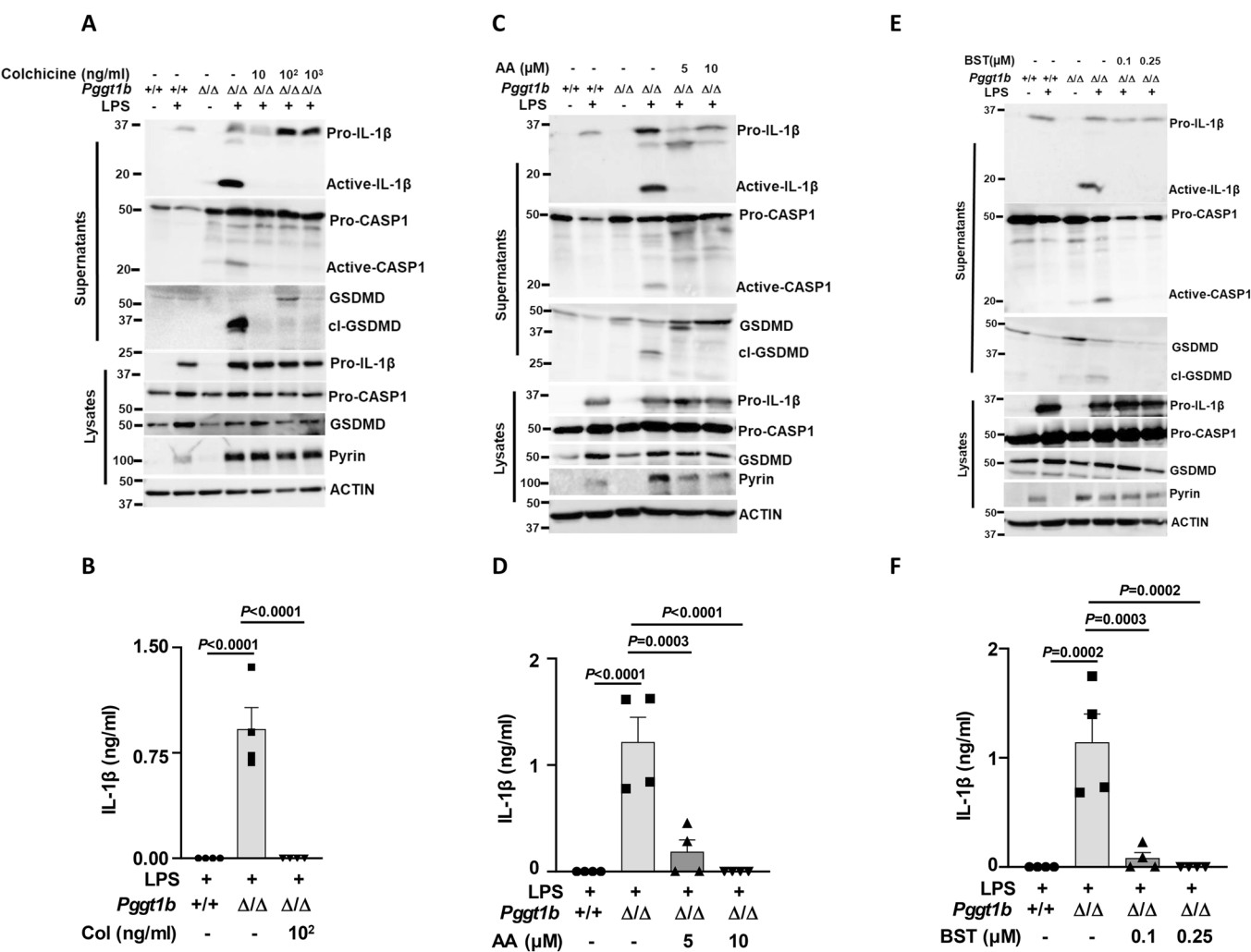

**Figure 6. Microtubule dynamics mediate Pyrin activation.**

(A) Western blots showing levels of pro- and bioactive-IL-1β, pro- and active-CASP1 and full length and cleaved (cl) GSDMD in supernatants and lysates from LPS-stimulated BMDMs isolated from $Pggt1b^{+/+}$ and $Pggt1b^{\Delta/\Delta}$ mice either pretreated or not with colchicine. (B) IL-1β cytokine levels, 8 h after LPS stimulation, in supernatants of BMDMs isolated from $Pggt1b^{+/+}$ ($n = 4$ biological replicates) and $Pggt1b^{\Delta/\Delta}$ ($n = 4$) mice either pretreated or not with colchicine. (C) Western blots showing levels of pro- and bioactive-IL-1β, pro- and active-CASP1 and full length and cleaved (cl) GSDMD in supernatants and lysates from LPS-stimulated BMDMs isolated from $Pggt1b^{+/+}$ and $Pggt1b^{\Delta/\Delta}$ mice either pretreated or not with arachidonic acid (AA). (D) IL-1β cytokine levels, 8 h after LPS stimulation, in supernatants of BMDMs isolated from $Pggt1b^{+/+}$ ($n = 4$ biological replicates) and $Pggt1b^{\Delta/\Delta}$ ($n = 4$) mice either pretreated or not with AA. (E) Western blots showing levels of pro- and bioactive-IL-1β, pro- and active-CASP1 and full length and cleaved (cl) GSDMD in supernatants and lysates from LPS-stimulated BMDMs isolated from $Pggt1b^{+/+}$ and $Pggt1b^{\Delta/\Delta}$ mice either pretreated or not with Bryostatin (BST). (F) IL-1β cytokine levels, 8 h after LPS stimulation, in supernatants of BMDMs isolated from $Pggt1b^{+/+}$ ($n = 4$ biological replicates) and $Pggt1b^{\Delta/\Delta}$ ($n = 4$) mice either pretreated or not with BST. Data are expressed as mean ± s.e.m. Significance between groups was calculated by one-way ANOVA and Dunnett's multiple comparison test. Western blots are representative of two independent experiments using biological replicates. Source data are available online for this figure.

we elucidate the role of unprenylated RAC1 in activating the Pyrin inflammasome via its interaction with the RAC1 effector IQGAP1. Importantly, inhibition of either Pyrin inflammasome activation or GSDMD-mediated pyroptosis effectively blocked IL-1β secretion by GGTase-I-deficient macrophages, thereby preventing joint inflammation in mice. These findings highlight the therapeutic potential of targeting Pyrin or GSDMD as a strategy for treating inflammasome-driven arthritis.

Our findings further establish GGTase-I as a critical regulator of Pyrin signaling in macrophages, functioning both as a suppressor of Pyrin protein expression and Pyrin inflammasome activation.

Defects in RHO GTPase prenylation, in conditions of GGTase-I deficiency, alleviate these restraints, promoting the interaction between Pyrin and the RAC1 effector IQGAP1, and driving Pyrin inflammasome hyperactivation. Consequently, this cascade leads to GSDMD-mediated pyroptosis and the release of bioactive IL-1β, ultimately resulting in the spontaneous development of erosive arthritis in GGTase-I-deficient mice.

Inhibition of RHO GTPases (GDP-bound) activates Pyrin through Pyrin dephosphorylation following exposure to Clostridial toxins (Gao et al, 2016; Park et al, 2016; Xu et al, 2014). Interestingly, a key phenotypic feature of GGTase-I knockout macrophages is the

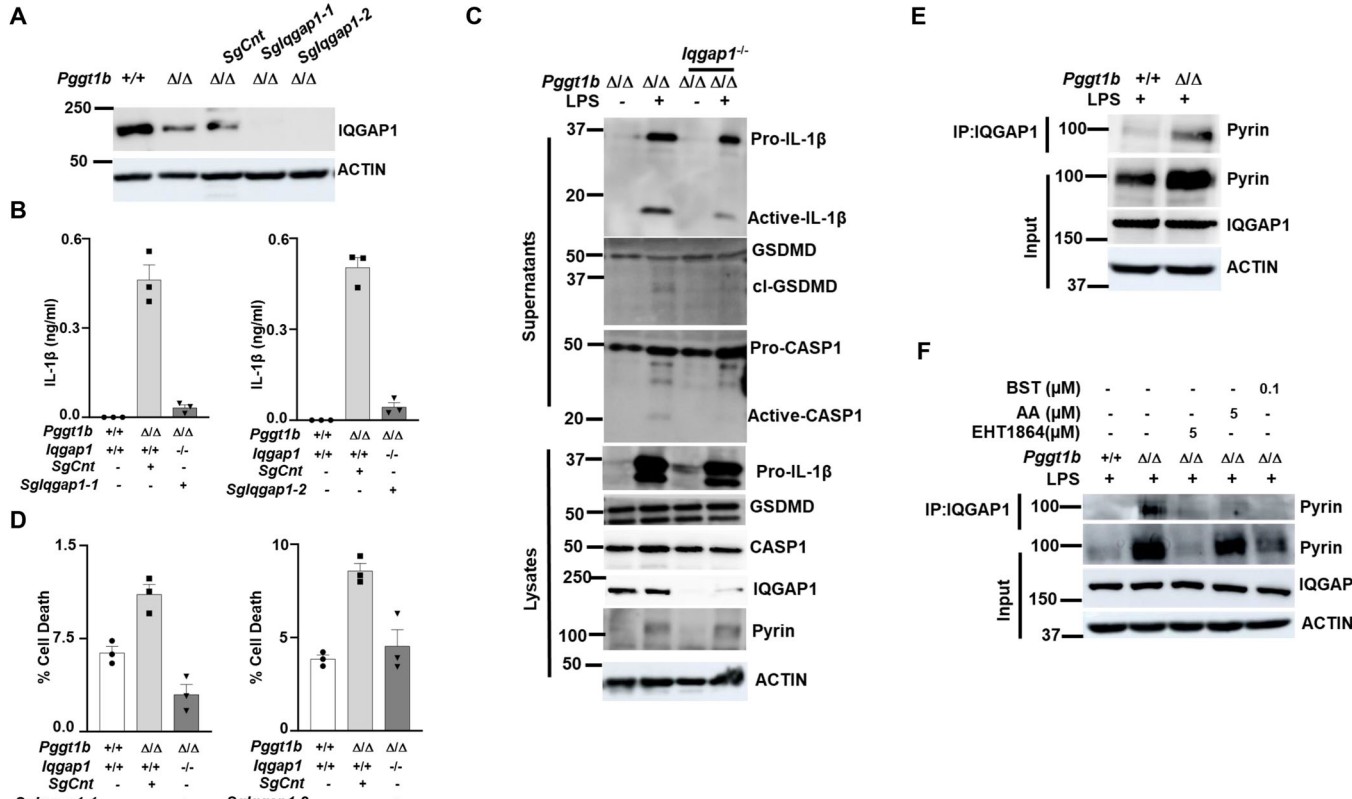

**Figure 7. IQGAP1 mediates Pyrin inflammasome activation in *Pggt1b*$^{\Delta/\Delta}$ macrophages.**

(A) Western blots showing levels of IQGAP1 and actin in cellular lysates isolated from *Pggt1b*$^{+/+}$ and *Pggt1b*$^{\Delta/\Delta}$-HoxB8 cells either transfected or not with a control single guide (sg) RNA or with IQGAP1-sgRNAs. (B) IL-1β cytokine levels in supernatants from *Pggt1b*$^{+/+}$ (n = 3), *Pggt1b*$^{\Delta/\Delta}$ (n = 3), and *Iqgap1*$^{-/-}$*Pggt1b*$^{\Delta/\Delta}$ (n = 3) macrophages 8 h after LPS stimulation. (C) Western blots showing levels of pro- and bioactive-IL-1β, pro- and cleaved (cl) CASP1 and full length and cleaved (cl) GSDMD in supernatants and lysates isolated from *Pggt1b*$^{+/+}$, *Pggt1b*$^{\Delta/\Delta}$ and *Iqgap1*$^{-/-}$*Pggt1b*$^{\Delta/\Delta}$ macrophages. (D) Percentage (%) of cell death, measured as LDH release in supernatants, 8 h after LPS stimulation of *Pggt1b*$^{+/+}$ (n = 3), *Pggt1b*$^{\Delta/\Delta}$ (n = 3), and *Iqgap1*$^{-/-}$*Pggt1b*$^{\Delta/\Delta}$ (n = 3) macrophages. (E) Immunoprecipitation (IP) of IQGAP1 in *Pggt1b*$^{+/+}$ and *Pggt1b*$^{\Delta/\Delta}$ macrophage lysates followed by Western blotting for Pyrin and IQGAP1. (F) Immunoprecipitation (IP) of IQGAP1 in *Pggt1b*$^{+/+}$ and *Pggt1b*$^{\Delta/\Delta}$ macrophage lysates either pretreated or not with EHT1864, arachidonic acid (AA), or Bryostatin (BST) followed by western blotting for Pyrin and IQGAP1. Actin was used as a loading control. Data are expressed as mean ± s.e.m. Western blots are representative of two independent experiments (except for **F**) using biological replicates. Source data are available online for this figure.

increased GTP-bound levels of RHO GTPases RAC1, RHOA, and CDC42 (Akula et al, 2019; Khan et al, 2011). In contrast to cells treated with Clostridial toxins—which directly and covalently inactivate RHO GTPases at the plasma membrane, leading to cytoskeletal disruption and proinflammatory responses (Xu et al, 2014)—GGTase-I deficiency may impair RHO GTPase function by preventing their prenylation and membrane localization. As a result, although RHO GTPases remain in their GTP-bound 'active' conformation, they are functionally inactive in macrophages due to their inability to associate with the plasma membrane. Therefore, while the mechanisms differ, the outcome of impaired prenylation of RHO family GTPases in GGTase-I-deficient conditions may resemble that observed with bacterial toxins, namely functional inactivation of RHO GTPase signaling leading to Pyrin inflammasome activation (Fig. EV7). We found that pharmacological inhibition of RAC1, but not RHOA, suppressed Pyrin activation in GGTase-I knockout macrophages, suggesting that RAC1 is responsible for the inflammatory phenotype observed in GGTase-I knockout mice. Collectively, this work identifies prenylation of RHO family GTPases as a checkpoint regulating Pyrin inflammasome activation.

We highlight IQGAP1 as a critical regulator bridging RAC1 to Pyrin inflammasome activation in conditions of GGTase-I deficiency. IQGAP1 was shown to bind nonprenylated RAC1 (Akula et al, 2019; Raulien et al, 2024) and to promote oxidative stress and mtDNA release, thereby activating the DNA sensor cGAS-STING and inducing GSDMD-mediated pyroptosis (An et al, 2023). Additionally, IQGAP1 has been shown to link inflammasome activation to GSDMD-dependent, ESCRT-mediated exosomal IL-1β release (Liao et al, 2023). Our data extend these findings by demonstrating that IQGAP1 recruits nonprenylated RAC1 to Pyrin, thereby promoting Pyrin inflammasome activation in GGTase-I–deficient macrophages, independently of changes in Pyrin expression. Notably, IQGAP1 has also been genetically linked to rheumatoid arthritis in humans (Leng et al, 2020), and our findings suggest that IQGAP1 may represent a promising therapeutic target for the treatment of Pyrin inflammasome-driven inflammatory disorders.

In vitro, Pyrin inflammasome activation and IL-1β secretion in GGTase-I-deficient macrophages require LPS. An outstanding question that remains is what specifically triggers Pyrin-

inflammasome-driven arthritis and bone inflammation in myeloid-specific GGTase-I-deficient mice? We speculate that sterile signals, including danger-associated molecular patterns (DAMPs) and mechanical strain, activate inflammatory responses in GGTase-I-deficient immune cells in the joints, driving musculoskeletal pathology. Multiple intracellular DAMPs are known to activate pattern recognition receptors and drive tissue inflammation when released in conditions of cell damage, including HMGB1, S100A8/A9, IL-1α, IL-33, ATP, uric acid, heat shock proteins, and mitochondrial DNA. In addition, extracellular matrix components can trigger immune activation and inflammation when released in damaged tissue, including fibronectin, tenascin-C, and hyaluronan (Danieli et al, 2022; Millerand et al, 2019; Taniguchi et al, 2018; Thiran et al, 2023). Alternatively, in addition to sterile signals, inflammation and arthritis in myeloid-specific GGTase-I-deficient mice may also be driven by microbiota-dependent mechanisms (Zaiss et al, 2021).

In summary, our studies highlight the regulatory role of the mevalonate pathway in controlling Pyrin inflammasome-driven inflammation and arthritis in mice. Non-prenylated RHO GTPases, prevalent in conditions of GGTase-I deficiency, induce Pyrin inflammasome activation and subsequent GSDMD-mediated pyroptosis and IL-1β release from macrophages. These findings also highlight the potential therapeutic value of targeting GSDMD and Pyrin directly, as well as the importance of RAC1 and its effector IQGAP1 as upstream regulators of the Pyrin inflammasome. Inhibiting these pathways may provide effective strategies to suppress Pyrin inflammasome activation and associated inflammatory responses.

# Methods

### Reagents and tools table

| Reagent/resource | Reference or source | Identifier or catalog number |
|---|---|---|
| **Experimental models** | | |
| *Pggt1b*<sup>fl</sup> mice | Khan et al, 2011 | |
| Lysozyme-M-Cre mice | Clausen et al, 1999 | |
| *Asc*<sup>−/−</sup> mice | Mariathasan et al, 2004 | |
| *Casp1*<sup>−/−</sup> mice | Van Gorp et al, 2016 | |
| *Gsdmd*<sup>−/−</sup> mice | Kayagaki et al, 2015 | |
| *Mefv*<sup>−/−</sup> mice | Van Gorp et al, 2016 | |
| *Il1r*<sup>−/−</sup> mice | Labow et al, 1997 | |
| *Myd88*<sup>−/−</sup> mice | Adachi et al, 1998 | |
| *Ripk1*<sup>D138N</sup> mice | Polykratis et al, 2014 | |
| *Casp8*<sup>−/−</sup> mice | Salmena et al, 2003 | |
| *Ripk3*<sup>−/−</sup> mice | Newton et al, 2004 | |
| *Tnfr*<sup>−/−</sup> mice | Pfeffer et al, 1993 | |
| *Nlrp3*<sup>−/−</sup> mice | Kanneganti et al, 2006 | |
| Primary cells (bone marrow-derived macrophages) | Derived from the respective mouse lines (in house) | |
| **Recombinant DNA** | | |
| Lenti-CRISPR-V2-GFP | Addgene | 82416 |
| **Antibodies** | | |
| Anti-IL-1β | R&D System | AF-401-NA |

| Reagent/resource | Reference or source | Identifier or catalog number |
|---|---|---|
| Anti-CASP1(p20) | Adipogen | AG-20B-0042-C100 |
| Anti-GSDMD | Abcam | ab209845 |
| Anti-Pyrin | Abcam | ab195975 |
| Anti-ASC | Millipore | 04-147 |
| Anti-MyD88 | Cell Signaling | 4283S |
| Anti-IQGAP1 | Cell Signaling | 2293S |
| Anti-CASP8 | Abnova | MAB3429 |
| Anti-RIPK3 | ProSci Incorporated | 2283 |
| Anti-RAC1 | Millipore | 05-389 |
| Anti-CDC42 | Cell Signaling | 2462S |
| Anti-RHOA | Cell Signaling | 2117S |
| Anti-NLRP3 | Adipogen | AG-20B-0014-C100 |
| Anti-Actin | Santa Cruz Biotechnology | sc-47778 |
| **Oligonucleotides and other sequence-based reagents** | | |
| Iqgap1 sgRNA1 | 5′-CACCGTAAAGCACGTCTTGGTACGT-3′ | IDT Technologies |
| Iqgap1 sgRNA2 | 5′-CACCGAGTCTACCTTGCCAAGCTA-3′ | IDT Technologies |
| *Pggt1b*<sup>fl</sup> | 5′-CCT GAA TGC AGA TCT GTG GA-3′ 5′-CCT ATG AAA GCA GCA CGA CA-3′ | |
| Lysozyme-M-Cre genotyping primers | 5′-CTT GGG CTG CCA GAA TTT CTC-3′ 5′-CCC AGA AAT GCC AGA TTA CG-3′ | |
| *Asc* genotyping primers | 5′-CTA GTT TGC TGG GGA AAG AAC-3′ 5′-CTA AGC ACA GTC ATT GTG AGC TCC-3′ 5′-AAG ACA ATA GCA GGC ATG CTG G-3′ | |
| *Casp1* genotyping primers | 5′-CGAGGGTTGGAGCTCAAGTTGACC-3′ 5′-CACTTTGACTTCTCTAAGGACAG-3′ | |
| *Gsdmd* genotyping primers | 5′-TTCCAATCCACAGCCTAAGAG-3′ 5′-AGCTCAATAAATAAACAAGAC-3′ 5′-CTGCTGTTGTTTCTACTACTC-3′ | |
| *Mefv* genotyping primers | 5′-CAGGCTACAGGGAGACAAGAA-3′ 5′-TCCTACCATTGCCACTGAGAG-3′ 5′-CAAAGGGAGCACAGACACTTC-3′ | |
| *Il1r* genotyping primers | 5′-CTCGTGCTTTACGGTATCGC-3′ 5′-GGTGCAACTTCATAGAGAGATGA-3′ 5′-TTCTGTGCATGCTGGAAAAC-3′ | |
| *Myd88* genotyping primers | 5′-TGG CAT GCC TCC ATC ATA GTT AAC C-3′ 5′-GTC AGA AAC AAC CAC CAC CAT GC-3′ 5′-ATC GCC TTC TAT CGC CTT CTT GAC G-3′ | |
| *Ripk1*<sup>D138N</sup> genotyping primers | 5′-TACCTTCTAACAAAGCTTTCC-3′ 5′-AATGGAACCACAGCATTGGC-3′ 5′-CCCTCGAAGAGGTTCACTAG-3′ | |
| *Casp8* genotyping primers | 5′-CCA GGA AAA GAT TTG TGT CTA-3′ 5′-GGC CTT CCT GAG TAC TGT CAC CTG-3′ | |
| *Ripk3* genotyping primers | 5′-CTG CTA ACC ATG TTC ATG CCT-3′ 5′-CCT GTT TTG CAC GTT CAC CG-3′ | |
| *Tnfr* genotyping primers | 5′-CTC TCT TGT GAT CAG CACT G-3′ 5′-CTG GAA GTG TGT CTC AC-3′ 5′-CCA AGC GAA ACA TCG CAT CGA GCG A-3′ | |
| *Nlrp3* genotyping primers | 5′-ACACCAGAATTTTGGGAGCCT-3′ 5′-TGGTATGACCGGACAGAGGG-3′ 5′-CCCTAGCTTTCAAAAAGAGTTGA-3′ | |
| **Chemicals, enzymes, and other reagents** | | |
| Recombinant Mouse IL-1RA | BioLegend | 769704 |
| 20 ng/ml GM-CSF | VIB Protein Core | |
| Polybrene | Sigma Aldrich | |

| Reagent/resource | Reference or source | Identifier or catalog number |
|---|---|---|
| RAC1 inhibitor-EHT1864 | Tocris | 3872 |
| RHOA inhibitor CCG1423 | Tocris | 5233 |
| Colchicine | Sigma Aldrich | C3915 |
| Arachidonic acid | Sigma Aldrich | 10931 |
| Bryostatin 1 | Tocris | 2383 |
| P38 Inhibitor SB203580 | Invivogen | SB203580 |
| GM-CSF | VIB Protein Core | |
| Beta-estradiol | Sigma Aldrich | E2758 |
| Hexadimethrine bromide | Sigma Aldrich | H9268 |
| **Software** | | |
| Graphpad Prism 10 | https://www.graphpad.com/features | |
| **Other** | | |
| ZEISS Axioscan 7 Microscope Slide Scanner | https://www.zeiss.com/microscopy/en/products/imaging-systems/axioscan-for-biology.html | |

## Mouse breeding and genotyping

The following mouse lines were used: $Pggt1b^{FL}$ (Khan et al, 2011), LysM-Cre (Clausen et al, 1999), $MefV^{-/-}$ (Van Gorp et al, 2016), $Casp1^{-/-}$ (Van Gorp et al, 2016), $Asc^{-/-}$ (Mariathasan et al, 2004), $Nlrp3^{-/}$ (Kanneganti et al, 2006), $Ripk1^{D138N}$ (Polykratis et al, 2014), $Gsdmd^{-/-}$ (Kayagaki et al, 2015), $Il-1r^{-/-}$ (Labow et al, 1997), $Tnfr^{-/-}$ (Pfeffer et al, 1993), $Myd88^{-/-}$ (Adachi et al, 1998), $Casp8^{-/-}$ (Salmena et al, 2003), and $Ripk3^{-/-}$ (Newton et al, 2004). All alleles were maintained on a C57BL/6 genetic background. Mice were housed in individually ventilated cages at the VIB Center for Inflammation Research, in a specific pathogen-free animal facility. Both male and female mice were used between 8 and 20 weeks of age. All experiments on mice were conducted according to institutional, national, and European animal regulations for animal testing and research. Animal protocols were approved by the VIB-Ghent University ethical review board (LP0010-2020).

## Isolation of BMDMs

BMDMs were obtained from bone marrow cells flushed from mouse femurs and tibia isolated from the respective mouse lines (both male and female) with ice-cold sterile DMEM medium, and cultured in DMEM high glucose medium with 50 ng/ml recombinant mouse M-CSF, 10% fetal bovine serum, 1% penicillin/streptavidin, 0.01% beta-mercaptoethanol, and glutamine. Fresh M-CSF was added on day 3, and the medium was refreshed on day 5. On day 7, cells were seeded at $1 \times 10^6$/well in 24-well plates for enzyme-linked immunosorbent assay (ELISA) analysis or at $2 \times 10^6$ cells/well in 12-well plates for immunoblot analysis.

## Generation of IQGAP1-deficient $Pggt1b^{\Delta/\Delta}$ macrophages

HoxB8-ER progenitors from $Pggt1b^{\Delta/\Delta}$ mice were generated based on the protocol described by Redecke et al (Redecke et al, 2013). In short, BMDMs were collected from the femur and tibia of 10-week-old mice by flushing with RPMI (ThermoFisher Scientific). Progenitor cells were purified by centrifugation over 3 ml Ficoll

Paque Plus (Sigma). Cells were resuspended in progenitor outgrowth medium (POM) consisting of RPMI (ThermoFisher Scientific) supplemented with 10% FCS (Bodinco), 1% penicillin/streptomycin (ThermoFisher Scientific), 20 ng/ml GM-CSF (VIB Protein Core), and 1 μM β-estradiol (Sigma). 250,000 cells per well were plated in 1 ml of POM in a fibronectin (Sigma) coated 12-well plate. Cells were infected by spinoculation ($1000 \times g$ for 1 h) with HoxB8-ER retroviral particles and in the presence of 0.8 μg/ml polybrene (Sigma). 3 ml of POM was added after spinfection. After 24 h, cells were collected, pelleted by centrifugation, and seeded in POM containing 1 μg/ml G418 (ThermoFisher Scientific) in 12-well suspension plates. Cells were split every 3–4 days until cell populations were stably expanding. To generate IQGAP1-deficient $Pggt1b^{\Delta/\Delta}$ HoxB8-ER progenitors, guide RNAs specific for IQGAP1 (5′-CACCGTAAAGCACGTCTTGGTACGT-3′ and 5′-CACCGAGTCTACCTTGCCAAGCTA-3′) were cloned into the lenti-CRISPR-V2-GFP plasmid. Lentiviral particles were produced in HEK293T cells after calcium phosphate transfection of the respective guide RNA plasmid together with gag/pol-encoding psPAX2 and the envelope-encoding pCMV-VSV-G helper plasmids. Lentiviral particles were added to $Pggt1b^{\Delta/\Delta}$-HoxB8 progenitor cells cultured in polybrene-containing medium (8 μg/ml) for 48 h. Seven days after, GFP-positive cells were sorted by FACS using BD FACS ARIAII sorter, and were cultured in medium containing GM-CSF (100 ng/ml) and beta-estradiol (5 μM) for 7–10 days. For differentiation to macrophages, $Pggt1b^{\Delta/\Delta}$-HoxB8 progenitor cells were washed twice with PBS to remove residual β-estradiol and plated at $10^6$ cells per 10 cm bacterial plate in 10 ml complete medium containing 100 ng/ml M-CSF for 7 days.

## Joint histology and arthritis assessment

Histology and analysis of mice joints was performed as described earlier (Gilis et al, 2019). Joints from 20-week-old mice were fixed in 4% formaldehyde for 24 h followed by 70% ethanol fixation. For the histopathologic evaluation of inflammation, mouse knees were fixed in 4% formaldehyde followed by decalcification in 5% formic acid. Paraffin-embedded sections of 7 μm were stained with hematoxylin and eosin, and pictures of the joints were captured by Axio. Arthritis scores were assessed by two investigators blinded to genotype for assessment of the severity of inflammation and damage to the joints. An arbitrary scale from 0 to 3 was used to indicate the arthritis index: 0—healthy joint, 1—mild immune cell infiltration in synovium and synovial tissue; 2—more pronounced infiltration of immune cells in synovium and synovial tissue; and 3—massive influx of inflammatory cells scattered throughout the synovium and synovial tissue.

## Cytokine detection

Macrophages ($10^6$/well) were plated in 24-well plates for ELISA experiments, and supernatants were isolated after treatment with or without LPS (20 ng/ml) for 8 h. For inhibitor experiments, macrophages were pre-treated for 30 min with RAC1 inhibitor (EHT1864, Tocris), RHOA inhibitor-CCG1423 (5233, Tocris), p38 Inhibitor (SB203580, Invivogen), or Colchicine (C3915, Sigma) followed by stimulation with LPS for 6 to 8 h. For IL-1β signaling inhibition experiments, macrophages were pretreated with recombinant Mouse IL-1RA (769704, BioLegend) for 3 h prior to stimulation with LPS. Cytokine concentrations in culture medium

**The paper explained**

**Problem**

Inflammatory arthritis occurs when the immune system mistakenly attacks the joints. This study investigates how disturbances in cholesterol-related metabolism might trigger such immune responses.

**Results**

Here, we studied the mevalonate pathway, which is responsible for synthesizing cholesterol and also helps regulate cellular signaling through a process called prenylation. When we blocked prenylation in immune cells by deleting the *Pggt1b* gene, key proteins (RHO GTPases) became mislocalized within the cell and triggered a strong inflammatory reaction. This response activated the Pyrin inflammasome, causing inflammatory cell death and the release of IL-1β, a major inflammatory signal. Deleting Pyrin pathway genes completely prevented arthritis in these mice.

**Impact**

The study links inflammatory arthritis to defects in the regulation of mevalonate signaling, highlighting Pyrin and IL-1β as possible treatment targets.

were determined by mIL-1β ELISA (88-7324-76, Thermofischer), according to the protocols provided by the manufacturer.

## Immunoprecipitation and immunoblotting

RAC1-GTP assays were performed with Active RAC1-GTP pulldown assay kit (16118, Thermo Fisher Scientific), and RHOA-GTP assays were performed with RHOA activation assay kit (BK036, Cytoskeleton). Proteins were isolated from supernatants using methanol:chloroform extraction procedure as described earlier (Akula et al, 2016). Cell lysates were prepared in E1A lysis buffer (250 mM NaCl, 50 mM Tris pH 7.4, 0.1% NP-40) containing a complete protease inhibitor cocktail (1:25) (Roche). For immunoprecipitation experiments, cells were lysed in NP-40 buffer (150 mM NaCl, 1% NP40, 10% glycerol, and 10 mM Tris–HCl pH 8). Immunoprecipitation experiments were performed with Protein G Sepharose 4 Fast Flow (17061801, Cytiva) kit, according to the manufacturer's instructions. Proteins were transferred to nitrocellulose membranes which were incubated with anti-IL-1β (AF-401-NA, R&D System, 1/500); anti-CASP1(p20) (AG-20B-0042-C100, Adipogen, 1/500); anti-GSDMD (ab209845, Abcam, 1/1000); anti-Pyrin (ab195975, Abcam, 1/1000); anti-ASC (04-147, Millipore, 1/1000); anti-MyD88 (4283S, Cell Signaling, 1/1000); anti-IQGAP1 (2293S, Cell Signaling, 1/1000); anti-CASP8 (MAB3429, Abnova, 1/1000); anti-RIPK3 (2283, ProSci Incorporated, 1/1000); anti-RAC1 (05-389, Millipore, 1/1000); anti-CDC42 (2462S, Cell Signaling, 1/1000); anti-RHOA (2117S, Cell Signaling, 1/1000); anti-NLRP3 (AG-20B-0014-C100, Adipogen, 1/1000); and anti-Actin (sc-47778, Santa Cruz Biotechnology, 1/10,000).

## Cell death assessment

Percentage cell death was assessed using a Cytotoxicity Assay (CytoTox 96® Non-Radioactive Cytotoxicity Assay, G1780, Promega) that quantitatively measures lactate dehydrogenase release

upon cell lysis in culture medium, according to the protocols provided by the manufacturer.

## Statistics

Data are represented as mean ± SEM. Statistical significance between experimental groups was assessed using one-way analysis of variance and Dunnett's multiple comparison test, and was considered significant when $P < 0.05$. Statistical details and sample size can be found in the figures and/or figure legends. Mice of both sexes were used in all experiments. No mice were excluded from analysis, and all in vivo analysis was done blindly to avoid subjective bias.

## Data availability

No primary datasets have been generated and deposited.

The source data of this paper are collected in the following database record: biostudies:S-SCDT-10_1038-S44321-025-00298-0.

## Peer review information

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

## Acknowledgements

We are grateful to Laetitia Bellen for animal care. MKA is a postdoctoral fellow
supported by a postdoctoral fellowship from the Vetenskapsrådet (2020-
06494) and Wenner-Gren Foundation-Sweden (WGF2021-0085). Research in
the van Loo lab is supported by VIB and by research grants from Ghent
University (BOF/24J/2021/052 and 01G00123), the FWO (3G090322,
3G0H2522), the Charcot Foundation, the Queen Elisabeth Medical Foundation,
the Belgian Foundation against Cancer (365L04523), and the FOREUM
Foundation for Research in Rheumatology. Research in the Lamkanfi lab is
supported by Ghent University (01G00123), and by research grants from the
FWO (GOI5722N, G017121N, G014221N) and European Research Council
(ERC-2022-PoC 101101075). Research in the Wullaert lab is supported by the
research grants 3G044718, 3G044818, G0A3422N, and G0A7O24N from the
FWO as well as the BOF UGent grant BOF.24Y.2019.0032.01.

## Author contributions

Murali K Akula: Conceptualization; Resources; Data curation; Formal analysis;
Funding acquisition; Validation; Investigation; Visualization; Methodology;
Writing—original draft; Writing—review and editing. Elisabeth Gilis:
Resources; Data curation; Formal analysis; Validation; Investigation;
Visualization. Pieter Hertens: Data curation; Formal analysis; Validation;
Investigation; Visualization. Lieselotte Vande Walle: Data curation; Formal
analysis; Validation; Investigation; Visualization. Mozes Sze: Data curation;
Formal analysis; Investigation. Julie Coudenys: Data curation; Formal analysis;
Investigation. Yunus Incik: Resources; Data curation; Methodology. Omar
Khan: Resources. Martin O Bergo: Resources. Dirk Elewaut: Resources;
Supervision; Methodology. Andy Wullaert: Resources; Validation; Writing—
original draft. Mohamed Lamkanfi: Resources; Supervision; Validation; Writing
—original draft; Writing—review and editing. Geert van Loo:
Conceptualization; Resources; Data curation; Formal analysis; Supervision;
Funding acquisition; Validation; Investigation; Methodology; Writing—original
draft; Project administration; Writing—review and editing.

Source data underlying figure panels in this paper may have individual
authorship assigned. Where available, figure panel/source data authorship is
listed in the following database record: biostudies:S-SCDT-10_1038-S44321-
025-00298-0.

## Disclosure and competing interests statement
The authors declare no competing interests.

# Expanded View Figures

**A**

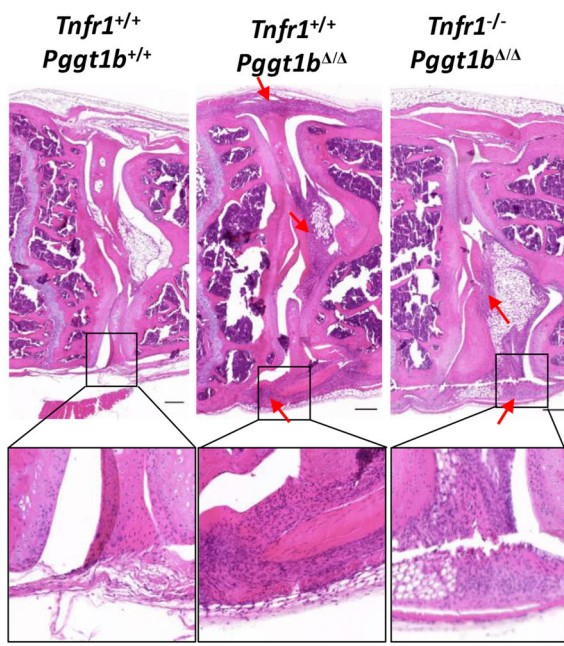

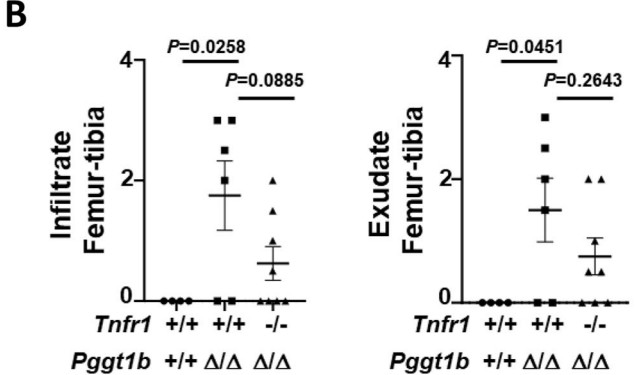

**Figure EV1. TNF does not drive arthritis development in *Pggt1b*^Δ/Δ mice.**

(A) Histological images of haematoxylin and eosin-stained knee joints of 20-week-old *Pggt1b*^+/+, *Pggt1b*^Δ/Δ, and *Tnfr*^−/−*Pggt1b*^Δ/Δ mice. Representative pictures are shown. Scalebar, 200 μm. Arrows depict inflammation, as shown by the accumulation of infiltrating leukocytes. (B) Histological scores for inflammation and exudate at the femur and tibia, each ranging from 0 (normal) to 3 (severely inflamed), of *Pggt1b*^+/+ ($n = 4$), *Pggt1b*^Δ/Δ ($n = 6$), and *Tnfr*^−/−*Pggt1b*^Δ/Δ ($n = 8$). Dots in the graphs indicate individual mice, and data are expressed as mean ± s.e.m. Significance between groups was calculated by one-way ANOVA and Dunnett's multiple comparison test. Source data are available online for this figure.

**A**

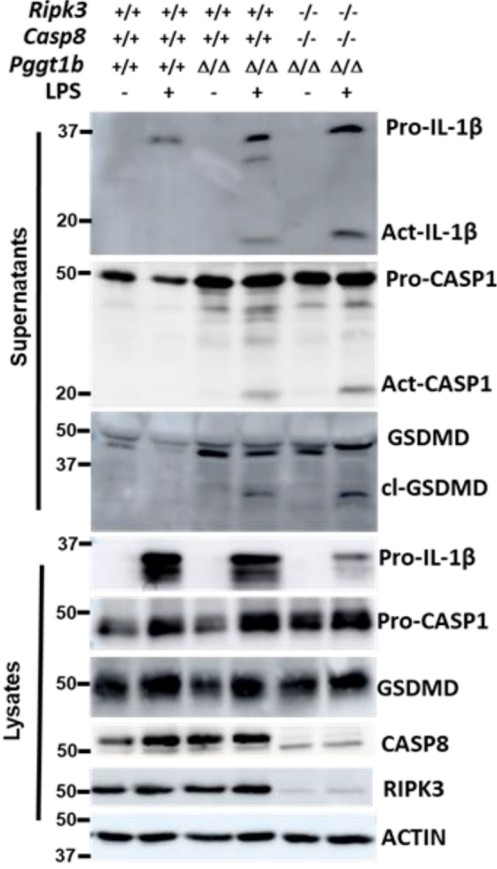

**C**

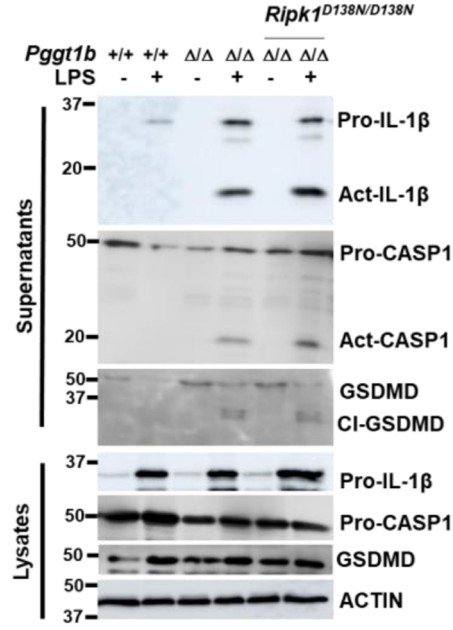

**B**

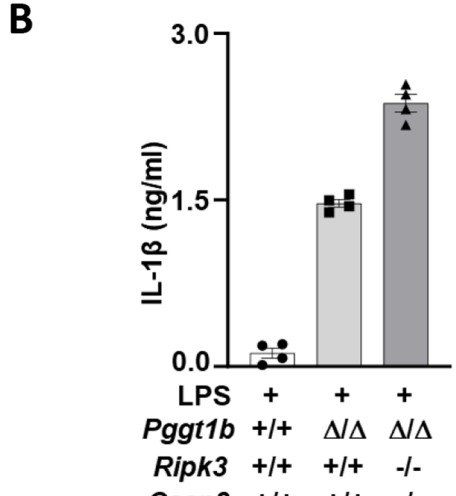

**D**

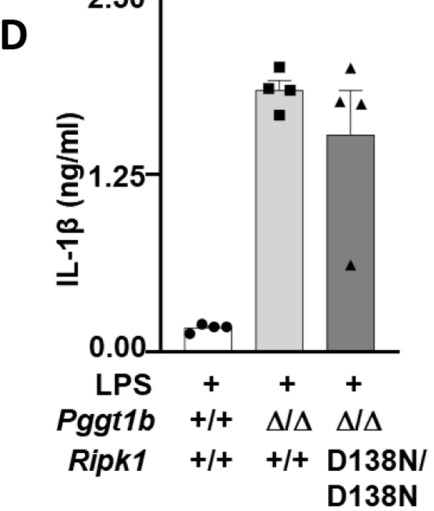

**Figure EV2.    Neither apoptosis nor necroptosis mediates inflammasome activation in *Pggt1b*$^{\Delta/\Delta}$ macrophages.**

(**A**) Western blots showing levels of pro- and active-IL-1β, pro- and active-CASP1 and full length and cleaved (cl) GSDMD in supernatants and lysates from BMDMs isolated from *Pggt1b*$^{+/+}$, *Pggt1b*$^{\Delta/\Delta}$, and *Casp8*$^{-/-}$*Ripk3*$^{-/-}$*Pggt1b*$^{\Delta/\Delta}$ mice either stimulated or not with LPS for 8 h. Actin was used as a loading control. (**B**) IL-1β levels in supernatants from BMDMs, isolated from *Pggt1b*$^{+/+}$ ($n = 4$ biological replicates), *Pggt1b*$^{\Delta/\Delta}$ ($n = 4$) and *Casp8*$^{-/-}$*Ripk3*$^{-/-}$*Pggt1b*$^{\Delta/\Delta}$ ($n = 4$) mice after treatment with LPS for 8 h. (**C**) Western blots showing levels of pro- and active-IL-1β, pro- and active-CASP1 and full length and cleaved (cl) GSDMD in supernatants and lysates from BMDMs isolated from *Pggt1b*$^{+/+}$, *Pggt1b*$^{\Delta/\Delta}$ and *Ripk1*$^{D138N/D138N}$*Pggt1b*$^{\Delta/\Delta}$ mice either stimulated or not with LPS for 8 h. Actin was used as a loading control. (**D**) IL-1β levels in supernatants from BMDMs, isolated from *Pggt1b*$^{+/+}$ ($n = 4$ biological replicates), *Pggt1b*$^{\Delta/\Delta}$ ($n = 4$), and *Ripk1*$^{D138N/D138N}$*Pggt1b*$^{\Delta/\Delta}$ ($n = 4$) mice after treatment with LPS for 8 h. Significance between groups was calculated by one-way ANOVA with Dunnett's multiple comparison test. Source data are available online for this figure.

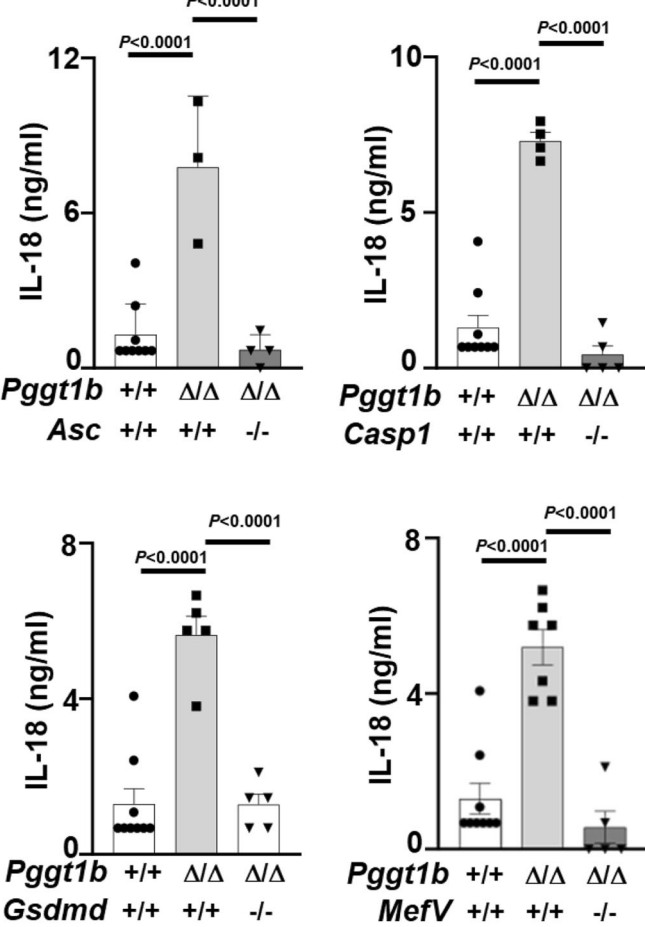

**Figure EV3.  Pyrin-mediated GSDMD signaling drives arthritis development in *Pggt1b*$^{\Delta/\Delta}$ mice.**

IL-18 cytokine levels in serum of *Pggt1b*$^{+/+}$ (*n* = 7 biological replicates), *Asc*$^{+/+}$*Pggt1b*$^{\Delta/\Delta}$ (*n* = 3), *Asc*$^{-/-}$*Pggt1b*$^{\Delta/\Delta}$ (*n* = 4), *Casp1*$^{+/+}$*Pggt1b*$^{\Delta/\Delta}$ (*n* = 4), *Casp1*$^{-/-}$*Pggt1b*$^{\Delta/\Delta}$ (*n* = 5), *Gsdmd*$^{+/+}$*Pggt1b*$^{\Delta/\Delta}$ (*n* = 5), *Gsdmd*$^{-/-}$*Pggt1b*$^{\Delta/\Delta}$ (*n* = 5), *Mefv*$^{+/+}$*Pggt1b*$^{\Delta/\Delta}$ (*n* = 7), and *Mefv*$^{-/-}$*Pggt1b*$^{\Delta/\Delta}$ (*n* = 5). Data are expressed as mean ± s.e.m. Significance between groups was calculated by one-way ANOVA and Dunnett's multiple comparison test. Source data are available online for this figure.

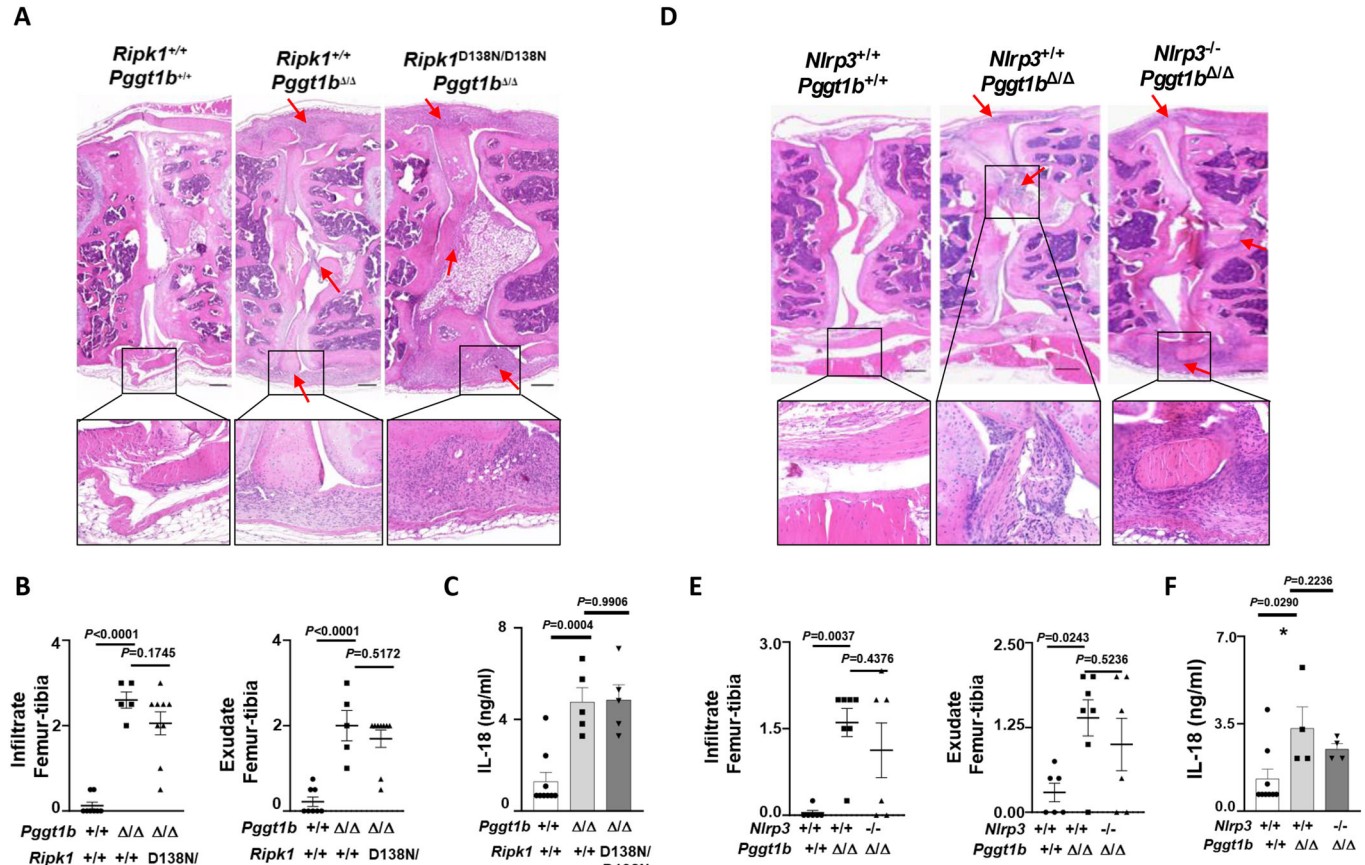

**Figure EV4. RIPK1 kinase and Nlrp3 do not drive arthritis development in Pggt1bΔ/Δ mice.**

(A) Histological images of haematoxylin and eosin-stained knee joints of Pggt1b+/+, Pggt1bΔ/Δ, and Ripk1D138N/D138NPggt1bΔ/Δ mice. Representative pictures are shown. Scalebar, 200 μm. Arrows depict inflammation. (B) Histological scores for inflammation and exudate at the femur and tibia, each ranging from 0 (normal) to 3 (severely inflamed), of Pggt1b+/+ (n = 9), Pggt1bΔ/Δ (n = 5), and Ripk1D138N/D138NPggt1bΔ/Δ (n = 9) mice. (C) IL-18 cytokine levels in serum of Pggt1b+/+ (n = 9 biological replicates), Ripk1+/+Pggt1bΔ/Δ (n = 5) and Ripk1D138N/D138NPggt1bΔ/Δ (n = 5) mice. (D) Histological images of haematoxylin and eosin-stained knee joints of Pggt1b+/+, Pggt1bΔ/Δ, and Nlrp3−/−Pggt1bΔ/Δ mice. Representative pictures are shown. Scalebar, 200 μm. Arrows depict inflammation. (E) Histological scores for inflammation and exudate at the femur and tibia, each ranging from 0 (normal) to 3 (severely inflamed), of Pggt1b+/+ (n = 6), Pggt1bΔ/Δ (n = 7), and Nlrp3−/−Pggt1bΔ/Δ (n = 6) mice. (F) IL-18 cytokine levels in serum of Pggt1b+/+ (n = 7 biological replicates), Nlrp3+/+Pggt1bΔ/Δ (n = 4) and Nlrp3−/−Pggt1bΔ/Δ (n = 4) mice. Data are expressed as mean ± s.e.m. Significance between groups was calculated by one-way ANOVA and Dunnett's multiple comparison test. Source data are available online for this figure.

**A**

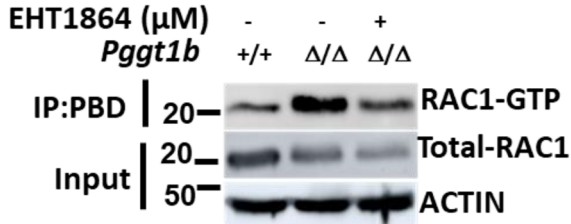

**B**

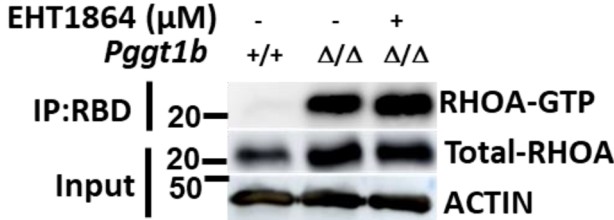

**C**

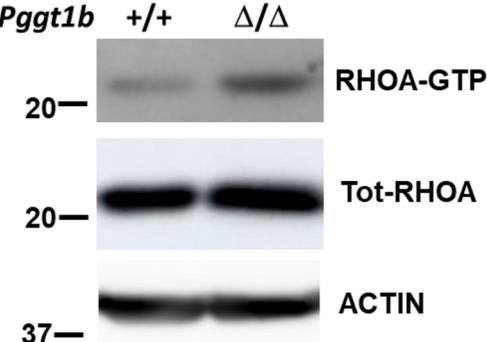

**Figure EV5.  RAC1 but not RHOA activates the Pyrin inflammasome in**
***Pggt1b^Δ/Δ* macrophages.**

(**A, B**) Immunoblots showing levels of total RAC1 and RAC1-GTP (**A**), and total
RHOA and RHOA-GTP (**B**) in LPS-stimulated BMDMs isolated from *Pggt1b*^+/+^
and *Pggt1b*^Δ/Δ^ mice either treated or not with EHT1864 for 8 h. (**C**) Immunoblots
showing levels of total RHOA and RHOA-GTP in lysates of *Pggt1b*^+/+^ and
*Pggt1b*^Δ/Δ^ BMDMs after treatment with LPS for 3 h. Actin was used as a loading
control. Source data are available online for this figure.

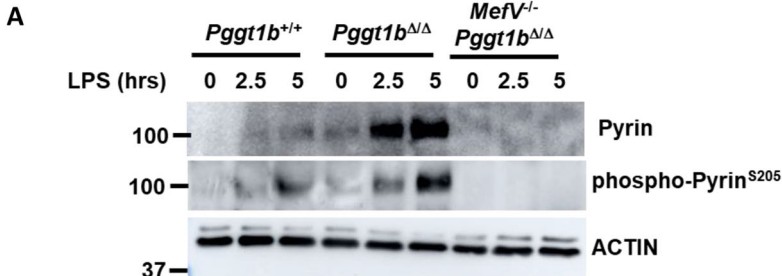

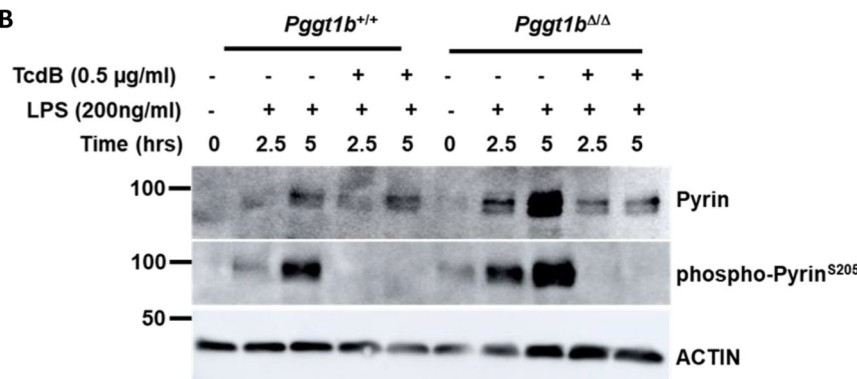

**Figure EV6.  *Pggt1b* deficiency does not regulate Pyrin phosphorylation levels directly.**

(A) Immunoblots showing Pyrin and phospho (S205)-Pyrin levels in lysates of LPS-treated *Pggt1b*[+/+], *Pggt1b*[Δ/Δ] and *Mefv*[−/−]*Pggt1b*[Δ/Δ] BMDMs. (B) Immunoblots showing Pyrin and phospho (S205)-Pyrin levels in lysates of LPS-treated *Pggt1b*[+/+] and *Pggt1b*[Δ/Δ] BMDMs in the presence or absence of TcdB (0.5 μg/ml). Actin was used as a loading control. Source data are available online for this figure.

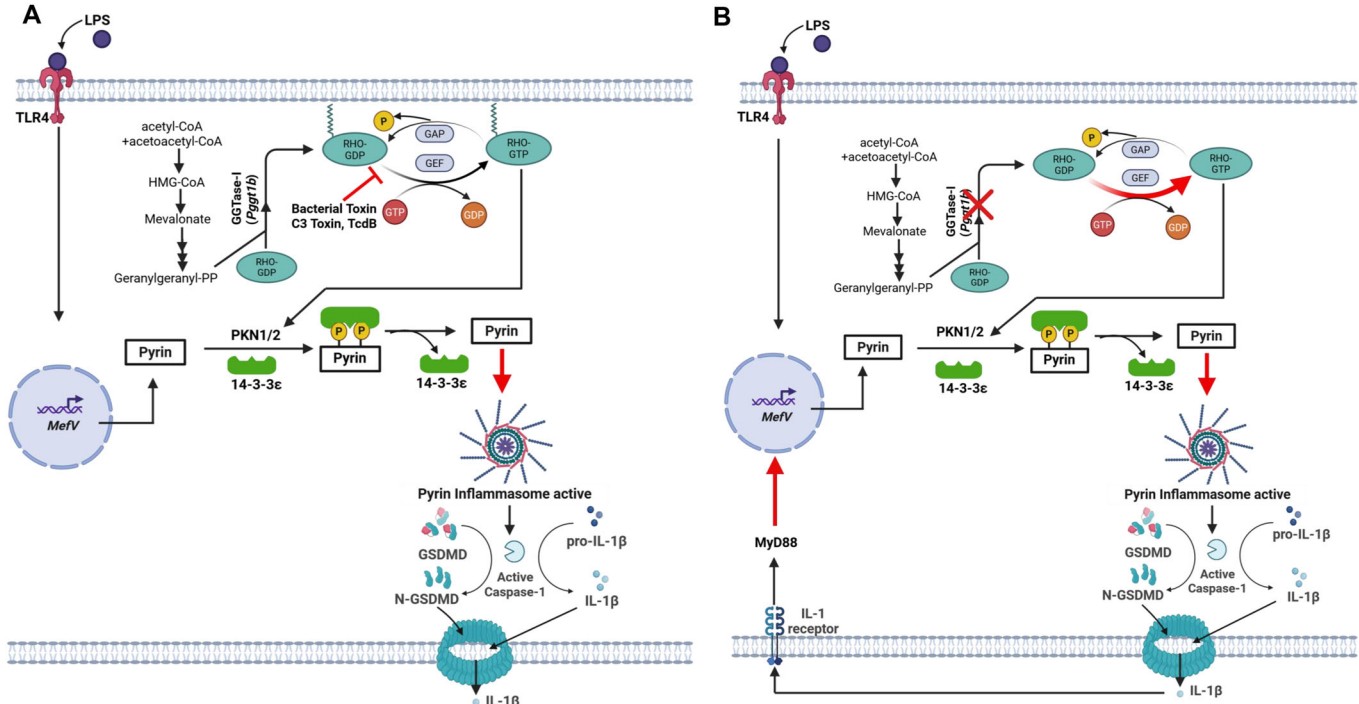

**Figure EV7.   Summarizing model of Pyrin inflammasome activation in toxin-treated and *Pggt1b*-deficient macrophages.**

(**A**) In untreated wildtype macrophages, geranylgeranyl pyrophosphate (GGPP)—a mevalonate pathway intermediate—facilitates the prenylation of RHO family GTPases, enabling their membrane localization and activation. Active RHO-GTP proteins engage effectors and activate kinases PKN1/2, which phosphorylate Pyrin and promote its binding to 14-3-3 proteins, thereby suppressing inflammasome activation. Bacterial toxins that inactivate RHO proteins prevent PKN1/2-mediated Pyrin phosphorylation, resulting in inflammasome assembly and IL-1β production. (**B**) In *Pggt1b*-deficient macrophages, impaired prenylation disrupts RHO protein membrane targeting and signaling. This leads to Pyrin inflammasome assembly with ASC and CASP1, cleavage of GSDMD and proIL-1β, and secretion of active IL-1β. Secreted IL-1β signals via the IL-1 receptor to upregulate *Mefv* transcription, thereby increasing Pyrin inflammasome activation. GTP guanine triphosphate, GDP guanine diphosphate, GEF guanine nucleotide exchange factors, GAP GTPases-activating proteins, TLR4 Toll-like receptor 4.

