## [Peer Review File · EMBO Molecular Medicine]

Pyrin inflammasome-driven erosive arthritis caused by unprenylated RHO GTPase signaling

Murali Akula, Elisabeth Gilis, Pieter Hertens, Lieselotte Vande Walle, Mozes Sze, Julie Coudenys, Yunus Incik, Omar Khan, Martin Bergö, Dirk Elewaut, Andy Wullaert, Mohamed Lamkanfi, and Geert van Loo

Corresponding authors: Geert van Loo (geert.vanloo@irc.vib-ugent.be) , Murali Akula (nagaa@irc.vib-ugent.be)

Review Timeline:

Submission Date:	31st Jan 25
Editorial Decision:	25th Feb 25
Revision Received:	26th May 25
Editorial Decision:	4th Jul 25
Revision Received:	10th Jul 25
Editorial Decision:	16th Jul 25
Revision Received:	25th Jul 25
Accepted:	30th Jul 25

Editor: Lise Roth

Transaction Report:

25th Feb 2025

Dear Prof. van Loo,

Thank you for the submission of your manuscript to EMBO Molecular Medicine. We have now heard back from the referees who reviewed your manuscript. As you will see from the reports below, they acknowledge the novelty and interest of the study and are overall supporting publication of your work pending appropriate revisions.

Addressing the reviewers' concerns in full will be necessary for further considering the manuscript in our journal, and acceptance of the manuscript will entail a second round of review. EMBO Molecular Medicine encourages a single round of revision only and therefore, acceptance or rejection of the manuscript will depend on the completeness of your responses included in the next, final version of the manuscript. For this reason, and to save you from any frustrations in the end, I would strongly advise against returning an incomplete revision.

We are expecting your revised manuscript within three months, if you anticipate any delay, please contact us.

We require:

4) A .docx formatted letter INCLUDING the reviewers' reports and your detailed point-by-point responses to their comments. As part of the EMBO Press transparent editorial process, the point-by-point response is part of the Review Process File (RPF), which will be published alongside your paper.

5) A complete author checklist, which you can download from our author guidelines (<https://www.embopress.org/page/journal/17574684/authorguide#submissionofrevisions>). Please insert information in the checklist that is also reflected in the manuscript. The completed author checklist will also be part of the RPF.

6) All Materials and Methods need to be described in the main text using our 'Structured Methods' format. According to this format, the Methods section includes a Reagents and Tools Table (listing key reagents, experimental models, software and relevant equipment and including their sources and relevant identifiers) followed by a Methods and Protocols section describing the methods, ideally using a step-by-step protocol format. The aim is to facilitate adoption of the methodologies across labs. Please download and fill our Reagents and Tools Table template (.docx), which you can find in our author guidelines:

<https://www.embopress.org/doi/10.15252/msb.20178071>

7) Please note that all corresponding authors are required to supply an ORCID ID for their name upon submission of a revised manuscript.

8) It is mandatory to include a 'Data Availability' section after the Materials and Methods. Before submitting your revision, primary datasets produced in this study need to be deposited in an appropriate public database, and the accession numbers and database listed under 'Data Availability'. Please remember to provide a reviewer password if the datasets are not yet public (see <https://www.embopress.org/page/journal/17574684/authorguide#dataavailability>).

9) For data quantification: please specify the name of the statistical test used to generate error bars and P values, the number (n) of independent experiments (specify technical or biological replicates) underlying each data point and the test used to calculate p-values in each figure legend. The figure legends should contain a basic description of n, P and the test applied. Graphs must include a description of the bars and the error bars (s.d., s.e.m.). Please provide exact p values.

10) Our journal encourages inclusion of *data citations in the reference list* to directly cite datasets that were re-used and obtained from public databases. Data citations in the article text are distinct from normal bibliographical citations and should directly link to the database records from which the data can be accessed. In the main text, data citations are formatted as follows: "Data ref: Smith et al, 2001" or "Data ref: NCBI Sequence Read Archive PRJNA342805, 2017". In the Reference list, data citations must be labeled with "[DATASET]". A data reference must provide the database name, accession number/identifiers and a resolvable link to the landing page from which the data can be accessed at the end of the reference. Further instructions are available at .

11) We replaced Supplementary Information with Expanded View (EV) Figures and Tables that are collapsible/expandable online. EV Figures should be cited as 'Figure EV1, Figure EV2' etc... in the text and their respective legends should be included in the main text after the legends of regular figures.

12) The paper explained: EMBO Molecular Medicine articles are accompanied by a summary of the articles to emphasize the major findings in the paper and their medical implications for the non-specialist reader. Please provide a draft summary of your article highlighting

13) Author contributions: CRediT has replaced the traditional author contributions section because it offers a systematic machine readable author contributions format that allows for more effective research assessment. Please remove the Authors Contributions from the manuscript and use the free text boxes beneath each contributing author's name in our system to add specific details on the author's contribution. More information is available in our guide to authors.

Please also suggest a visual abstract to illustrate your article as a PNG file 550 px wide x 300-600 px high. A cropped portion of this image will serve as thumbnail for the table of content on our webpage.

16) As part of the EMBO Publications transparent editorial process initiative (see our Editorial at <http://embomolmed.embopress.org/content/2/9/329>), EMBO Molecular Medicine will publish online a Review Process File (RPF) to accompany accepted manuscripts.

In the event of acceptance, this file will be published in conjunction with your paper and will include the anonymous referee reports, your point-by-point response and all pertinent correspondence relating to the manuscript. Let us know whether you agree with the publication of the RPF and as here, if you want to remove or not any figures from it prior to publication. Please note that the Authors checklist will be published at the end of the RPF.

I look forward to receiving your revised manuscript.

Yours sincerely,

Lise Roth

**** Reviewer's comments ****

Referee #1 (Remarks for Author):

This study explores the role of geranylgeranyl pyrophosphate and geranylgeranyl transferase I (GGTase-I) in inflammatory arthritis, revealing a pathway where myeloid-specific Pgg1b deletion leads to RAC1 hyperactivation, Pyrin inflammasome activation, and subsequent pyroptosis and IL-1 β release. The authors provide evidence using in vivo genetic models to dissect the molecular mechanisms underlying these processes, highlighting the potential pro-inflammatory effects of prolonged statin use.

The genetic experiments presented are convincing. However, several technical points require clarification and further investigation to solidify the proposed mechanisms.

Comments:

GGTase-I deficiency leads to a rapid increase in Pyrin expression within 2 hours. The authors should clarify whether this upregulation is due to a positive feedback loop from IL-1 β production as part of a priming effect (signal 1). Additionally, it would be helpful to determine if this rapid induction occurs in the presence of simvastatin. The observation that RAC1 inhibition reduces this effect suggests an upstream regulatory role that warrants further exploration.

The use of RhoA inhibitors in Figure 5A is unclear. Since RhoA inhibition can activate Pyrin, it is crucial to assess whether GGTase inhibition directly affects RhoA activity. Supplementary Figure 4 suggests that GGTase deficiency may lock RhoA in its active GTP-bound state. The authors should investigate how this affects PKN-mediated Pyrin phosphorylation and its interaction with 14-3-3. Experiments using bone marrow-derived macrophages (BMDMs) could provide further clarity.

While PKC activators were used to address this point, the connection between the observed effects and a direct RAC1-PKC pathway remains speculative. Since PKC inhibition is known to suppress Pyrin, this does not necessarily imply that RAC1 functions through PKC. This hypothesis should be tested by directly monitoring Pyrin phosphorylation and 14-3-3 binding.

The study demonstrates that RAC1 activation and IQGAP1 deficiency contribute to inflammasome activation following isoprenoid pathway inhibition. However, the broader relevance of these findings remains unclear. The authors should assess how RAC1 inhibition and IQGAP1 deficiency impact Pyrin activation under relevant conditions, such as TcdB exposure for example.

Previous studies (doi: 10.1038/ni.3457) have established that simvastatin can activate Pyrin through RhoA inhibition. The link between RAC1 and RhoA in the present study remains ambiguous. Given that RAC1 is known to inhibit RhoA (and vice versa), the authors should consider whether RAC1 activation indirectly modulates RhoA activity, contributing to Pyrin activation.

Minor Comment:

Supplementary Figure 2 (inflammasome pathway) should be moved into Figure 2 for better visualization, while the necroptosis

pathway could be relegated to the supplementary figures.

Referee #2 (Remarks for Author):

Geranylation - the conjugation of geranylgeranyl (a non-sterol intermediate of the mevalonate pathway) onto target proteins - has previously been implicated in the regulation of RHO family proteins. Because RHO family proteins are regulators of inflammatory responses, initially inhibiting the geranylation of RHO family proteins was considered as a possible strategy to suppress inflammatory diseases. Contrary to expectations, follow-up findings discovered that myeloid-specific deletion of *Pggt1b*, the gene encoding geranylgeranyl transferase, results in spontaneous and severe erosive arthritis characterized by enhanced inflammatory responses and IL-1 β secretion. The mechanisms behind this erosive arthritis have been unclear. In this study, Akula MK and colleagues demonstrate that inflammation and erosive arthritis in mice with myeloid-specific *Pggt1b* deficiency are caused by hyperactive RAC1 signaling and downstream activation of the Pysin inflammasome, which in turn induces GSDMD-dependent macrophage pyroptosis and secretion of IL-1 β . Genetic deletion of *Il1r1*, *ASC*, *Casp1*, *Gsdmd*, or *Pysin*, but not *Nlrp3*, rescued the arthritic phenotype caused by myeloid-restricted *Pggt1b* deficiency. Further, they show that *Pggt1b* deficiency is associated with hyperactive RAC1 signaling, which induces Pysin inflammasome activation by recruiting Pysin to the RAC1 effector IQGAP1. Finally, they demonstrate that prolonged treatment with the clinically licensed statin drug simvastatin, which interferes with prenylation, can also trigger Pysin activation, pyroptosis, and IL-1 β release in LPS-primed macrophages through a mechanism that depends on hyperactive RAC1 signaling, thus illustrating the potentially detrimental pro-inflammatory mechanisms of prolonged statin use.

This is a very interesting study with well-conducted experiments that support the authors' conclusions. I support its publication as it is.

Referee #3 (Remarks for Author):

The paper of Akula et al describes a mechanistic study proposing a novel pathway in the pathogenesis of erosive arthritis. The study makes ambivalent impression with very strong positive sides but also strong negative sides. On the positive side are well defined and novel hypothesis, very strong mechanistic aspect (and this is especially attractive), very good design encompassing investigating ALL elements of the proposed pathway, a combination of in vitro and in vivo studies supporting each other, sound methodology. On the negative side are relaxed attitude toward requirements for replication and quantitation, and to corroborate key findings with an independent method, neglecting alternative explanations and poor scientific writing (see the last comment for the specifics on that point). There is no need to add much to what is good, as to what is wrong, below are the specific comments.

Specific comments:

1. Most of the data, and the most important data, are presented as Western blots - without quantitation and biological and technical replicates shown. This is a serious limitation, confidence in such data is low.
2. p.5, Fig.1. It would be beneficial if morphological data were corroborated with an independent readout, e.g. any blood inflammatory marker. It would also be very informative if plasma levels of IL-1b were measured.
3. p. 6-7, Fig.2,3. The evidence that pyroptosis contributes, and perhaps significantly to the IL-1b release in this model, and even works in the absence of *Nlrp3*, are very convincing. That, however, does not mean that there is no contribution of *Nlrp3* inflammasome altogether in *Pggt1b*^{+/+} animals; this should be tested and quantitated.
4. p.8. Fig.5. I request evidence, or at least references, on the specificity of EHT1864 and CCG1423.
5. p.9. Statement that colchicine "selectively inhibits Pysin activation" is incorrect, all microtubules around the cells will be disrupted affecting most intracellular trafficking and making it difficult to attribute the effects exclusively to Pysin.
6. Fig 7 and associated text. As the authors themselves mention, IQGAP1 interacts with many GTPases and consequently its deletion would affect many of them impacting many aspects of recycling machinery, if not intracellular traffic in general. What are the "unequivocal" evidence that the observed effects are due to Pysin inflammasome activation, and not due to, just as an example, reduced LPS signalling via TLR?
7. [REDACTED: referee comments and author response on unpublished data].
8. Last, but not least. The way many figures are constructed and described in the text is very confusing and requires a lot of effort to understand. I explain what I mean using mainly Fig. 2 as an example, but this applies to most figures: 1. Panels in the figure

are usually placed in alphabetical order, here it's rather random. 2. With bar graphs, there is no mention if statistic is based on technical or biological replicates. 3. Y-axis labelling is cryptic: concentration of IL1 where, after what time, what does cell death mean, you are measuring LDH, a marker. 4. Figs 1 and 4 show joints stained with HE and text states that there is a difference in development of arthritis, see arrows, no further comment. Well, perhaps for an expert in morphology of joints it is obvious what to look for, the problem is, I'm not, and need some guidance, like look at the accumulation of macrophages here and there, just few comments on what elements of morphology were assessed. 5. Fig. 5 is described as follows: "Pharmacological inhibition of RAC1, but not RHOA, abrogated LPS-induced IL-1 β processing and secretion, as well as caspase-1 and GSDMD cleavage in Pgg1b Δ/Δ macrophages, in a dose-dependent manner (Fig. 5A-C)". This is not a description what is shown, this is a conclusion, and the reader is left to their own devices to figure out where to look and what to look for. I can continue this list for another page. This attitude is unacceptable, the authors should think about how a reader would read the figures.

Dear Editor,

We would like to thank the Reviewers for their instructive and considerate comments. We are happy that the Reviewers acknowledge the importance of our findings, however also recognize a number of caveats that need to be addressed.

We have put in a considerable effort to meet all of the points that were raised by the Reviewers by conducting a series of novel experiments and by adapting the manuscript text, as appropriate.

Point-by-point response to

Referee #1 :

"The genetic experiments presented are convincing. However, several technical points require clarification and further investigation to solidify the proposed mechanisms."

We are grateful to this Reviewer for his/her positive assessment of our work and for his/her constructive criticisms, which have helped to further strengthen the proposed disease mechanisms. We have fully addressed the Reviewer's comments as detailed below:

"GGTase-I deficiency leads to a rapid increase in Pysin expression within 2 hours. The authors should clarify whether this upregulation is due to a positive feedback loop from IL-1 β production as part of a priming effect (signal 1). Additionally, it would be helpful to determine if this rapid induction occurs in the presence of simvastatin. The observation that RAC1 inhibition reduces this effect suggests an upstream regulatory role that warrants further exploration."

We thank the Reviewer for this important suggestion, which has helped to gain further insights into the signalling mechanisms leading to increased Pysin expression. To address the reviewer's suggestion, we stimulated WT and *Pggt1b* knockout macrophages with LPS in the presence or absence of recombinant mouse interleukin-1 receptor (IL-1R) Antagonist (50ng/ml) to block IL-1R signalling (rebuttal Figure 1A below). Consistent with our previous results, *Pggt1b* knockout macrophages showed increased Pysin expression levels compared to their controls in response to LPS. However, blocking IL-1R signalling significantly diminished Pysin levels in *Pggt1b* knockout macrophages. These results suggest that aberrant production of IL-1 β released from GGTase-I (*Pggt1b*) knockout macrophages contributes importantly to the priming signal and further increases Pysin expression levels as part of a positive feedback loop. These results were included in the revised manuscript as a new panel Figure 3C.

To address the Reviewer's additional question regarding simvastatin treatment and its effect on Pysin expression levels, we pretreated macrophages with or without simvastatin for 16 hours followed by a kinetic analysis of Pysin expression following LPS stimulation. As in *Pggt1b* knockout macrophages, we observed increased Pysin expression in simvastatin-treated macrophages compared to their controls in response to LPS (rebuttal Figure 1B below). These new results are now shown in Suppl. Figure EV7I of the revised manuscript.

Fig. 1. A. Immunoblots showing Pyrin levels in lysates of LPS-treated *Pggt1b^{+/+}* and *Pggt1b^{Δ/Δ}* macrophages, in the presence or absence of recombinant mouse interleukin-1 receptor (IL-1R) Antagonist (50 ng/ml). Actin was used as a loading control. B. Immunoblots showing Pyrin levels in lysates of control macrophages either untreated or treated with simvastatin for 24 hours, followed by LPS stimulation for varying durations.

“The use of RhoA inhibitors in Figure 5A is unclear. Since RhoA inhibition can activate Pyrin, it is crucial to assess whether GGTase inhibition directly affects RhoA activity.

We apologize if the presentation of the results with the RhoA inhibitors in Figure 5A was not sufficiently clear. The reasoning behind this was that previous work (Akula *et al*, 2019; Khan *et al*, 2011) demonstrated that hyperactivation of RHO family proteins, particularly RAC1-GTP, RHOA-GTP and CDC42-GTP, represents a key phenotypic feature of GGTase-I knockout macrophages. However, how these effects contributed to Pyrin signaling in GGTase-deficient macrophages was not clear. Please note that in contrast to cells treated with Clostridial toxins—which directly and covalently inactivate RHO GTPases at the plasma membrane, leading to cytoskeletal disruption and proinflammatory responses (Xu *et al*, 2014)—GGTase-I deficiency may impair RHO GTPase function by preventing their prenylation and membrane localization. As a result, although RHO GTPases remain in their GTP-bound ‘active’ conformation, they are functionally inactive in macrophages due to their inability to associate with the plasma membrane. Therefore, while the mechanisms differ, the outcome of impaired prenylation of RHO family GTPases in GGTase-I deficient conditions may resemble that observed with bacterial toxins, namely functional inactivation of RHO GTPase signaling leading to Pyrin inflammasome activation. This concept is now addressed in the Discussion section of the revised manuscript.

Interestingly, we found that pharmacological inhibition of RAC1-GTP, but not RHOA-GTP, suppressed Pyrin activation in GGTase-I knockout macrophages, suggesting that RAC1-GTP is responsible for the inflammatory phenotype observed in GGTase-I knockout mice. To test the effect of GGTase inhibition on RHOA activity directly, as requested by the Reviewer, we now performed an experiment where we pulled down RHOA-GTP from LPS-treated *Pggt1b* knockout and wildtype BMDMs, confirming RHOA hyperactivation (RHOA-GTP) in *Pggt1b* knockout macrophages upon LPS stimulation (rebuttal Figure 2 below). These new data are included in the revised manuscript (Suppl. Figure EV5C).

Fig. 2: Hyperactivation of RHOA in GGTase-I knockout macrophages. Immunoblots showing levels of RHOA-GTP and total-RHOA in lysates of *Pggt1b^{+/+}* and *Pggt1b^{Δ/Δ}* BMDMs after treatment with LPS for three hours. Actin was used as a loading control.

Collectively, this work identifies prenylation of RHO family GTPases as a checkpoint regulating Pyrin inflammasome activation.

Supplementary Figure 4 suggests that GGTase deficiency may lock RhoA in its active GTP-bound state. The authors should investigate how this affects PKN-mediated Pyrin phosphorylation and its interaction with 14-3-3. Experiments using bone marrow-derived macrophages (BMDMs) could provide further clarity.”

Previous study demonstrated that PKN1 and PKN2 phosphorylate Pyrin, which subsequently binds to 14-3-3 proteins, regulatory proteins that in turn block the Pyrin inflammasome (Park *et al.*, 2016). As suggested by the Reviewer, we have analyzed PKN-mediated Pyrin phosphorylation levels and the interaction between Pyrin and 14-3-3 in BMDMs of wildtype and *Pggt1b* knockout macrophages. We observe that the levels of phospho-Ser205 increase concomitantly with increasing LPS-induced Pyrin expression levels in *Pggt1b* knockout macrophages (rebuttal Figure 3A below), suggesting that despite increasing total Pyrin protein expression levels, the ratio of Pyrin phosphorylation relative to total Pyrin protein expression in control and *Pggt1b* knockout macrophages is relatively similar. Treatment with the Clostridial toxin TcdB, however, suppresses Pyrin phosphorylation in wildtype macrophages, as shown in previous studies (Xu *et al.*, 2014). Similar observations are made in *Pggt1b* knockout macrophages where all Pyrin becomes unphosphorylated following treatment with TcdB (rebuttal Figure 3B below). Consistent with this finding, we observed Pyrin binding with 14-3-3ε in both *Pggt1b* knockout macrophages and control macrophages relative to their Pyrin expression levels (rebuttal Figure 3C below). These results suggest that *Pggt1b* deficiency does not regulate Pyrin phosphorylation levels directly. These new data are included in the revised manuscript (new Suppl. Figure EV6).

Fig. 3: A-B. Immunoblots showing Pyrin and phospho (S205)-Pyrin levels in lysates of LPS-treated *Pggt1b*^{+/+}, *Pggt1b*^{Δ/Δ} and *MefV*^{-/-}*Pggt1b*^{Δ/Δ} BMDMs in the presence or absence of TcdB (0.5 μg/ml). C. Immunoprecipitation of 14-3-3ε in BMDM lysates followed by immunoblotting for Pyrin and 14-3-3ε. Direct immunoblots were performed on the same lysates to identify the total Pyrin and 14-3-3ε levels. Actin was used as a loading control.

“While PKC activators were used to address this point, the connection between the observed effects and a direct RAC1-PKC pathway remains speculative. Since PKC inhibition is known to suppress Pyrin, this does not necessarily imply that RAC1 functions through PKC. This hypothesis should be tested by directly monitoring Pyrin phosphorylation and 14-3-3 binding.”

We kindly refer the Reviewer to our response above.

“The study demonstrates that RAC1 activation and IQGAP1 deficiency contribute to inflammasome activation following isoprenoid pathway inhibition. However, the broader relevance of these findings remains unclear. The authors should assess how RAC1 inhibition and IQGAP1 deficiency impact Pyrin activation under relevant conditions, such as TcdB exposure for example.”

Unfortunately we were not able to generate RAC1 nor IQGAP1 deficient macrophages to directly address Reviewer’s question. We successfully deleted IQGAP1 in HoxB8-immortalized *Pggt1b* knockout macrophages using CRISPR-Cas9 technology (Figure 7), however did not manage to generate IQGAP1 deficient *Pggt1b*^{+/+} macrophages.

However, consistent with published findings from Feng Shao’s lab and others demonstrating that Pyrin activation in response to Clostridial toxins such as TcdB is triggered by RHO GTPase inhibition (Xu *et al.*, 2014), we propose that impaired prenylation may also contribute to the Pyrin inflammasome-mediated inflammatory phenotype observed in GGTase-I-deficient cells or cells treated with simvastatin. As discussed above, disruption of RHO GTPase prenylation—whether due to GGTase-I deficiency or statin treatment—prevents their membrane localization and thereby impairs their downstream signaling, thereby allowing Pyrin activation to proceed unchecked. Our discussion of this mechanistic concept of Pyrin inflammasome activation is now more elaborately discussed in the revised Discussion section of the manuscript.

“Previous studies (doi: 10.1038/ni.3457) have established that simvastatin can activate Pyrin through RhoA inhibition. The link between RAC1 and RhoA in the present study remains ambiguous. Given that RAC1 is known to inhibit RhoA (and vice versa), the authors should consider whether RAC1 activation indirectly modulates RhoA activity, contributing to Pyrin activation.”

We thank the Reviewer for raising an important point, and we fully agree that addressing this will help to clarify the complex interactions between RAC1 and RHOA in our settings. The cited study (Park *et al.*, 2016) showed that simvastatin treatment can activate Pyrin in wildtype macrophages, which is consistent with our results. However, as far as we can tell, this study didn’t show any direct evidence that this occurs through RHOA inhibition.

In our studies, we found hyperactivation of RAC1 and RHOA in macrophages after treatment with simvastatin, rather than RHOA inhibition. Furthermore, RAC1 inhibitor (EHT1864) treatment did not appear to modulate RHOA-GTP levels in simvastatin-treated macrophages (rebuttal Figure 4 below). Similar results were observed in *Pggt1b* knockout macrophages after treatment with LPS (Suppl. Figure EV5C of the revised manuscript).

Fig. 4. Immunoblots showing levels of RHOA-GTP and total-RHOA in lysates of control and simvastatin-treated macrophages after treatment with EHT-1864 (5 μ M) for four hours. Actin was used as a loading control.

However, in contrast to cells stimulated with TcdB or other Clostridial toxins—which directly and covalently inactivate RHO GTPases at the membrane leading to cytoskeletal disruption and proinflammatory responses—we propose that *Pggt1b* deficiency or treatment with simvastatin prevents RHO GTPase function by abolishing their prenylation and membrane localization. Consequently, although RHOA remains in its GTP-bound ‘active’ conformation, it is functionally inactive due to its inability to localize to the membrane. Therefore, while the mechanisms differ, the outcome resembles that observed with bacterial toxins: RHO GTPase inactivation. Based on this concept, RHOA may still contribute to Pyrin activation under conditions of *Pggt1b* deficiency or simvastatin treatment. This concept is now discussed more extensively in the revised manuscript discussion.

“Minor Comment: Supplementary Figure 2 (inflammasome pathway) should be moved into Figure 2 for better visualization, while the necroptosis pathway could be relegated to the supplementary figures.”

In response to the Reviewer's suggestion, we have revised the manuscript by relocating certain figure panels from the main figures to the supplementary figures, and vice versa.

Referee #2 :

“This is a very interesting study with well-conducted experiments that support the authors' conclusions. I support its publication as it is.”

We thank this Reviewer for his/her positive assessment and shared excitement about the importance and rigor of this work.

Referee #3 :

“The paper of Akula et al describes a mechanistic study proposing a novel pathway in the pathogenesis of erosive arthritis. The study makes ambivalent impression with very strong positive sides but also strong negative sides. On the positive side are well defined and novel hypothesis, very strong mechanistic aspect (and this is especially attractive), very good design encompassing investigating ALL elements of the proposed pathway, a combination of in vitro and in vivo studies supporting each other, sound methodology. On the negative side are relaxed attitude toward requirements for replication and quantitation, and to corroborate key findings with an independent method, neglecting alternative explanations and poor scientific writing (see the last comment for the specifics on that point). There is no need to add much to what is good, as to what is wrong, below are the specific comments.”

We thank the Reviewer's comments on the positive aspects of our study, and hope that the additional data sets and more elaborate discussion of our findings as presented in the revised manuscript fully and satisfactorily address the outstanding critiques of the Reviewer. In the following, we provide a point-by-point reply to the individual comments.

1. "Most of the data, and the most important data, are presented as Western blots - without quantitation and biological and technical replicates shown. This is a serious limitation, confidence in such data is low."

Western blotting is a standard methodology in the Inflammasome signaling field, as it uniquely allows to visualize core proteolytic events that are associated with inflammasome activation and downstream IL1 β production and secretion in macrophages. Hence, nearly all publications in the inflammasome field incorporate this as a core methodology to demonstrate inflammasome activation. These proteolytic events include the cleavage of pro-caspase-1 into the mature protease caspase-1, the processing of the caspase-1 substrate pro-IL1 β into the mature cytokine IL1 β , and the cleavage of full-length Gasdermin-D (GSDMD) into the cleaved GSDMD fragment that translocates to the membrane and allows pore formation and IL1 β release. Although protein bands in Western blots can be quantified by densitometry, we believe this is here redundant since the differences between conditions are clearly distinguishable by eye.

In addition to our Western blots, all our studies are complemented by ELISA assays to measure IL1 β cytokine levels secreted in the culture media. In some cases we also include assays to measure the degree of pyroptosis induction making use of a commercially available cytotoxicity assay based on lactate dehydrogenase (LDH) release in the culture media. Moreover, the *in vitro* studies are complemented by *in vivo* analyses through histology on knee joints to address the degree of arthritis pathology in GGTase-I deficient mice. In response to the Reviewers' comments, we have further included ELISA measurements of serum IL-18 levels in GGTase-I-deficient mice crossed to inflammasome-deficient mice. Together, we believe our use of well-established and complementary technical approaches provide robust support for our scientific conclusions.

As indicated in the figure legends, all data are representative of two independent experiments using different biological samples, *viz.* BMDMs isolated from different mice.

2. "p.5, Fig.1. It would be beneficial if morphological data were corroborated with an independent readout, e.g. any blood inflammatory marker. It would also be very informative if plasma levels of IL-1b were measured."

We thank the Reviewer for his/her suggestion and now complemented the histology data with an independent readout, *viz.* IL18 cytokine levels in serum that had been collected from mice from the different mouse lines. We also tried to measure IL-1 β levels in the serum of these mice, however all values are below the detection limit of the ELISA assay. IL-1 β is notoriously difficult to detect in serum samples from mice under non-infectious conditions, hence IL-18, which is also a product of inflammasome activation but is more stable and less prone to rapid degradation than IL-1 β in blood samples, often serves as a useful surrogate marker for inflammasome-related processes. To support this claim, we refer to a study investigating serum cytokine biomarkers in FMF (Familial Mediterranean fever) (Koga *et al*, 2016), caused by gain-of-function mutations in the Pyrin-encoding *MEFV* gene, and a study on CAPS (cryopyrin-associated periodic syndrome) (Lachmann *et al*, 2009), caused by gain-of-function mutations in *NLRP3*. Although both autoinflammatory diseases are caused by

uncontrolled caspase-1 activation and IL-1 β secretion, these studies demonstrate that IL-1 β is virtually undetectable in human plasma.

As shown in Figure 5 of this rebuttal, IL18 cytokine levels, measured in the serum of these mice, nicely align with the arthritis assessment based on histology. These new figure panels have been added to the revised manuscript in Suppl. Figures EV3 and EV4.

Fig. 5. IL-18 cytokine levels in serum of $Pggt1b^{+/+}$ (n = 7), $Asc^{+/+}Pggt1b^{\Delta/\Delta}$ (n = 3), $Asc^{-/-}Pggt1b^{\Delta/\Delta}$ (n = 4), $Casp1^{+/+}Pggt1b^{\Delta/\Delta}$ (n = 4), $Casp1^{-/-}Pggt1b^{\Delta/\Delta}$ (n = 5), $Gsdmd^{+/+}Pggt1b^{\Delta/\Delta}$ (n = 5), $Gsdmd^{-/-}Pggt1b^{\Delta/\Delta}$ (n = 5), $MefV^{+/+}Pggt1b^{\Delta/\Delta}$ (n = 7), $MefV^{-/-}Pggt1b^{\Delta/\Delta}$ (n = 5), $Myd88^{+/+}Pggt1b^{\Delta/\Delta}$ (n = 9), $Myd88^{-/-}Pggt1b^{\Delta/\Delta}$ (n = 6), $Nlrp3^{+/+}Pggt1b^{\Delta/\Delta}$ (n = 4), $Nlrp3^{-/-}Pggt1b^{\Delta/\Delta}$ (n = 4), $Ripk1^{+/+}Pggt1b^{\Delta/\Delta}$ (n = 5), $Ripk1^{D138N/D138N}Pggt1b^{\Delta/\Delta}$ (n = 5), $Tnfr1^{+/+}Pggt1b^{\Delta/\Delta}$ (n = 4), $Tnfr1^{-/-}Pggt1b^{\Delta/\Delta}$ (n = 5). One way ANOVA analysis. *P < 0.05, **P < 0.01, ***P < 0.001, ****P < 0.0001.

3. “p. 6-7, Fig.2,3. The evidence that pyroptosis contributes, and perhaps significantly to the IL-1 β release in this model, and even works in the absence of *Nlrp3*, are very convincing. That, however, does not mean that there is no contribution of *Nlrp3* inflammasome altogether in *Pggt1b*^{+/+} animals; this should be tested and quantitated.”

We now performed a control experiment using BMDMs from wild-type and *Nlrp3* knockout mice (both wildtype for *Pggt1b*). As expected and shown in Figure 6 of this rebuttal, stimulation of control cells (*Nlrp3*^{+/+}) with LPS alone does not induce IL1 β secretion. For this you need a second signal such as Nigericin, which induces strong IL1 β secretion in wildtype BMDMs, which is completely absent in *Nlrp3* knockout BMDMs.

Fig. 6. IL-1 β levels in supernatants of *Nlrp3*^{+/+} and *Nlrp3*^{-/-} macrophages 8 hours after LPS stimulation, in presence or absence of Nigericin.

4. "p.8. Fig.5. I request evidence, or at least references, on the specificity of EHT1864 and CCG1423."

We used the small molecule EHT1864 to inhibit RAC1-GTP in GGTase-I knockout macrophages (Supp. Fig. EV5). EHT1864 was reported to selectively inhibit RAC1 and its downstream signalling pathway, but not RHOA (Onesto *et al*, 2008; Shutes *et al*, 2007). Following these initial validation studies, many other studies have used EHT1864 to target RAC1 signalling (Dutting *et al*, 2015; Rosenblatt *et al*, 2011; Sun *et al*, 2021; Zhang *et al*, 2025).

To inhibit RHOA signalling, we have used the small molecule CCG1423. CCG1423 has been validated as a selective inhibitor of RHOA in multiple studies (Becker *et al*, 2024; Evelyn *et al*, 2007; Li *et al*, 2024; Rosen *et al*, 2023; Zhang *et al.*, 2025).

However, we acknowledge that small molecule inhibitors can have cross-reactivity with other GTPase family proteins.

5. "p.9. Statement that colchicine "selectively inhibits Pyn activation" is incorrect, all microtubules around the cells will be disrupted affecting most intracellular trafficking and making it difficult to attribute the effects exclusively to Pyn."

We agree with the Reviewer that colchicine affects all microtubules. Although colchicine is used to inhibit Pyn activation, we have revised the sentence and removed the term 'selectively' for greater accuracy. In addition to the colchicine data, our study includes data with *MefV* knockout cells and mice, confirming that Pyn activation is responsible for IL1 β production and inflammation in GGTase-I knockout mice and statin-treated cells.

6. "Fig 7 and associated text. As the authors themselves mention, IQGAP1 interacts with many GTPases and consequently its deletion would affect many of them impacting many aspects of recycling machinery, if not intracellular traffic in general. What are the "unequivocal" evidence that the observed effects are due to Pyn inflammasome activation, and not due to, just as an example, reduced LPS signalling via TLR?"

We demonstrated that IQGAP1 deletion reduces Pyn inflammasome activation and the release of IL1 β in GGTase-I knockout macrophages (Figure 7). However, IQGAP1 deletion did not affect pro-IL1 β levels, as shown in Figure 7C of the manuscript, indicating that IQGAP1 knockout macrophages respond normally to LPS stimulation. Since we agree with the Reviewer that IQGAP1 through its interaction with multiple GTPases may regulate many aspects of intracellular signaling and cytoskeletal dynamics, we now toned down our statement of "unequivocal" evidence.

7. [REDACTED: referee comments and author response on unpublished data].

8. *“Half of the discussion is repeating presenting the results in a plain language, rather than discussing implications of these interesting findings, which are many.”*

Although we start our discussion with a short overview of our findings, we also elaborate on the implications of our findings by relating them to the existing literature, by discussing the possible reasons for Pypin inflammasome priming *in vivo* in GGtase-I deficient mice that develop arthritis, and by discussing the issue of the inflammatory side effects of statins. The controversy about the anti- vs. pro-inflammatory effects of statins, as discussed above, is now more extensively discussed in the revised manuscript. We now also discussed the nature

of the RHO GTPases responsible for the activation of the Pysin inflammasome in our models (*Pggt1b* deficiency and simvastatin treatment) vs. the bacterial toxins, as discussed in our feedback to the questions of Reviewer 1.

9. *“Last, but not least. The way many figures are constructed and described in the text is very confusing and requires a lot of effort to understand. I explain what I mean using mainly Fig. 2 as an example, but this applies to most figures: 1. Panels in the figure are usually placed in alphabetical order, here it's rather random. 2. With bar graphs, there is no mention if statistic is based on technical or biological replicates. 3. Y-axis labelling is cryptic: concentration of IL1 where, after what time, what does cell death mean, you are measuring LDH, a marker. 4. Figs 1 and 4 show joints stained with HE and text states that there is a difference in development of arthritis, see arrows, no further comment. Well, perhaps for an expert in morphology of joints it is obvious what to look for, the problem is, I'm not, and need some guidance, like look at the accumulation of macrophages here and there, just few comments on what elements of morphology were assessed. 5. Fig. 5 is described as follows: "Pharmacological inhibition of RAC1, but not RHOA, abrogated LPS-induced IL-1 β processing and secretion, as well as caspase-1 and GSDMD cleavage in *Pggt1b Δ/Δ macrophages, in a dose-dependent manner (Fig. 5A-C)". This is not a description what is shown, this is a conclusion, and the reader is left to their own devices to figure out where to look and what to look for. I can continue this list for another page. This attitude is unacceptable, the authors should think about how a reader would read the figures.”**

We apologize if the presentation of the data was not sufficiently clear. We have tried to improve on this in the revised version of the manuscript.

1. Although we attempted to organize all figure panels alphabetically, in certain instances, layout considerations necessitated alternative arrangements, which only minimally deviate from their normal alphabetical order.
2. All statistical analyses accompanying bar graphs are based on biological replicates. This has now been explicitly clarified in the legend of the first bar graph, where the number of mice is specified.
3. The figure legends have now been clarified in greater detail in the materials and methods section of the revised manuscript. IL-1 β cytokine levels are measured 6 to 8 h after LPS stimulation in supernatants of BMDMs. Percentage cell death is measured using a Cytotoxicity Assay (CytoTox 96[®] Non-Radioactive Cytotoxicity Assay, Promega) that quantitatively measures lactate dehydrogenase (LDH), a stable cytosolic enzyme that is released upon cell lysis in culture medium.
4. Higher magnification insets have been added to all histology images (Figure 1, Figure 4, Supplementary Figure EV1, and Supplementary Figure EV4), along with additional explanatory text to support our claims.
5. We have tried to improve the narrative and provide a more elaborate discussion of our findings and their broader impact on the field to help the reader follow our experimental observations and discuss their impact from a broader biomedical viewpoint.

References :

- Akula MK, Ibrahim MX, Ivarsson EG, Khan OM, Kumar IT, Erlandsson M, Karlsson C, Xu X, Brisslert M, Brakebusch C *et al* (2019) Protein prenylation restrains innate immunity by inhibiting Rac1 effector interactions. *Nat Commun* 10: 3975
- Becker IC, Barrachina MN, Lykins J, Camacho V, Stone AP, Chua BA, Signer RAJ, Machlus KR, Whiteheart SW, Roweth HG *et al* (2024) Inhibition of RhoA-mediated secretory autophagy in megakaryocytes mitigates myelofibrosis in mice. *bioRxiv*
- Bjorkhem-Bergman L, Lindh JD, Bergman P (2011) What is a relevant statin concentration in cell experiments claiming pleiotropic effects? *Br J Clin Pharmacol* 72: 164-165
- Chamani S, Kooshkaki O, Moossavi M, Rastegar M, Soflaei SS, McCloskey AP, Banach M, Sahebkar A (2023) The effects of statins on the function and differentiation of blood cells. *Arch Med Sci* 19: 1314-1326
- Dutting S, Heidenreich J, Cherpokova D, Amin E, Zhang SC, Ahmadian MR, Brakebusch C, Nieswandt B (2015) Critical off-target effects of the widely used Rac1 inhibitors NSC23766 and EHT1864 in mouse platelets. *J Thromb Haemost* 13: 827-838
- Evelyn CR, Wade SM, Wang Q, Wu M, Iniguez-Lluhi JA, Merajver SD, Neubig RR (2007) CCG-1423: a small-molecule inhibitor of RhoA transcriptional signaling. *Mol Cancer Ther* 6: 2249-2260
- Healy A, Berus JM, Christensen JL, Lee C, Mantsounga C, Dong W, Watts JP, Jr., Assali M, Ceneri N, Nilson R *et al* (2020) Statins Disrupt Macrophage Rac1 Regulation Leading to Increased Atherosclerotic Plaque Calcification. *Arterioscler Thromb Vasc Biol* 40: 714-732
- Henriksbo BD, Lau TC, Cavallari JF, Denou E, Chi W, Lally JS, Crane JD, Duggan BM, Foley KP, Fullerton MD *et al* (2014) Fluvastatin causes NLRP3 inflammasome-mediated adipose insulin resistance. *Diabetes* 63: 3742-3747
- Henriksbo BD, Tamrakar AK, Phulka JS, Barra NG, Schertzer JD (2020) Statins activate the NLRP3 inflammasome and impair insulin signaling via p38 and mTOR. *Am J Physiol Endocrinol Metab* 319: E110-E116
- Khan OM, Ibrahim MX, Jonsson IM, Karlsson C, Liu M, Sjogren AK, Olofsson FJ, Brisslert M, Andersson S, Ohlsson C *et al* (2011) Geranylgeranyltransferase type I (GGTase-I) deficiency hyperactivates macrophages and induces erosive arthritis in mice. *J Clin Invest* 121: 628-639
- Kiener PA, Davis PM, Murray JL, Youssef S, Rankin BM, Kowala M (2001) Stimulation of inflammatory responses in vitro and in vivo by lipophilic HMG-CoA reductase inhibitors. *Int Immunopharmacol* 1: 105-118
- Koga T, Migita K, Sato S, Umeda M, Nonaka F, Kawashiri SY, Iwamoto N, Ichinose K, Tamai M, Nakamura H *et al* (2016) Multiple Serum Cytokine Profiling to Identify Combinational Diagnostic Biomarkers in Attacks of Familial Mediterranean Fever. *Medicine (Baltimore)* 95: e3449
- Kuijk LM, Mandey SH, Schellens I, Waterham HR, Rijkers GT, Coffey PJ, Frenkel J (2008) Statin synergizes with LPS to induce IL-1beta release by THP-1 cells through activation of caspase-1. *Mol Immunol* 45: 2158-2165
- Lachmann HJ, Lowe P, Felix SD, Rordorf C, Leslie K, Madhoo S, Wittkowski H, Bek S, Hartmann N, Bosset S *et al* (2009) In vivo regulation of interleukin 1beta in patients with cryopyrin-associated periodic syndromes. *J Exp Med* 206: 1029-1036
- Li Y, Zuo C, Wu X, Ding Y, Wei Y, Chen S, Lu X, Xu J, Liu S, Zhou G *et al* (2024) FBXL8 inhibits post-myocardial infarction cardiac fibrosis by targeting Snail1 for ubiquitin-proteasome degradation. *Cell Death Dis* 15: 263
- Liao YH, Lin YC, Tsao ST, Lin YC, Yang AJ, Huang CT, Huang KC, Lin WW (2013) HMG-CoA reductase inhibitors activate caspase-1 in human monocytes depending on ATP release and P2X7 activation. *J Leukoc Biol* 93: 289-299
- Onesto C, Shutes A, Picard V, Schweighoffer F, Der CJ (2008) Characterization of EHT 1864, a novel small molecule inhibitor of Rac family small GTPases. *Methods Enzymol* 439: 111-129
- Park YH, Wood G, Kastner DL, Chae JJ (2016) Pyrin inflammasome activation and RhoA signaling in the autoinflammatory diseases FMF and HIDS. *Nat Immunol* 17: 914-921

- Rosen E, Mangukiya HB, Elfineh L, Stockgard R, Krona C, Gerlee P, Nelander S (2023) Inference of glioblastoma migration and proliferation rates using single time-point images. *Commun Biol* 6: 402
- Rosenblatt AE, Garcia MI, Lyons L, Xie Y, Maiorino C, Desire L, Slingerland J, Burnstein KL (2011) Inhibition of the Rho GTPase, Rac1, decreases estrogen receptor levels and is a novel therapeutic strategy in breast cancer. *Endocr Relat Cancer* 18: 207-219
- She J, Tuerhongjiang G, Guo M, Liu J, Hao X, Guo L, Liu N, Xi W, Zheng T, Du B *et al* (2024) Statins aggravate insulin resistance through reduced blood glucagon-like peptide-1 levels in a microbiota-dependent manner. *Cell Metab* 36: 408-421 e405
- Shutes A, Onesto C, Picard V, Leblond B, Schweighoffer F, Der CJ (2007) Specificity and mechanism of action of EHT 1864, a novel small molecule inhibitor of Rac family small GTPases. *J Biol Chem* 282: 35666-35678
- Skinner OP, Jurczyk J, Baker PJ, Masters SL, Rios Wilks AG, Clearwater MS, Robertson AAB, Schroder K, Mehr S, Munoz MA *et al* (2019) Lack of protein prenylation promotes NLRP3 inflammasome assembly in human monocytes. *J Allergy Clin Immunol* 143: 2315-2317 e2313
- Sun J, Gaidosh G, Xu Y, Mookhtiar A, Man N, Cingaram PR, Blumenthal E, Shiekhattar R, Goka ET, Nimer SD *et al* (2021) RAC1 plays an essential role in estrogen receptor alpha function in breast cancer cells. *Oncogene* 40: 5950-5962
- Ward NC, Watts GF, Eckel RH (2019) Statin Toxicity. *Circ Res* 124: 328-350
- Xu H, Yang J, Gao W, Li L, Li P, Zhang L, Gong YN, Peng X, Xi JJ, Chen S *et al* (2014) Innate immune sensing of bacterial modifications of Rho GTPases by the Pyrin inflammasome. *Nature* 513: 237-241
- Zhang T, Wang Y, Nie X, Chen X, Jin Y, Sun L, Yang R, Wang J, Xu W, Song T *et al* (2025) ENKD1 modulates innate immune responses through enhanced geranylgeranyl pyrophosphate synthase activity. *Cell Rep* 44: 115397

4th Jul 2025

Dear Prof. van Loo,

Thank you for submitting your revised manuscript to EMBO Molecular Medicine, and please accept my apologies for the delay in responding. As I mentioned in an earlier email, we received the reports from referees #1 and #3, and given their divergent opinions, we discussed the matter further with the referees and an external expert advisor.

As agreed with you, we would like to invite you to make the following revisions to your manuscript:

- Remove the statin-related data.
- Provide the source data for all WB replicates (no quantification required).

Furthermore, please address the following editorial matters:

1/ Manuscript text:

- Please remove the yellow highlights and only keep in track changes mode any new modification in the text.
- Please provide up to 5 keywords.
- Methods:
 - o Cell material(s) should be listed in the Reagent and Tools table.
 - o Mice: please indicate gender and age at time of experiments
 - o Antibodies: provide dilutions/concentrations for all antibodies.
 - o Statistics: please provide statements on sample size, inclusion/exclusion criteria, blinding and randomization.
- Please provide a Disclosure statement and competing interests statement: Please review the policy <https://www.embopress.org/competing-interests> and update your competing interests if necessary.
- Acknowledgements: please make sure that the complete list of funders that need to be mentioned is entered into our system. Currently, a number of funders are only listed in the Acknowledgements and should be entered in our system via More Funders option (not in the Comments box): FWO (3G090322, 3G0H2522), the Charcot Foundation, the Queen Elisabeth Medical Foundation, the Belgian Foundation against Cancer (365L04523), and the FOREUM Foundation for Research in Rheumatology; the FWO (GOI5722N, G017121N, G014221N) and European Research Council (ERC-2022-PoC 101101075); grants 3G044718, 3G044818, G0A3422N and G0A7O24N from the FWO as well as the BOF UGent grant BOF.24Y.2019.0032.01.

2/ Figures:

- Please provide individual production quality figure files as .eps, .tif, .jpg (one file per figure) for the main and EV figures. For guidance, download the 'Figure Guide PDF' (<https://www.embopress.org/page/journal/17574684/authorguide#figureformat>).
- Please upload all WB at higher resolution, as they are currently overpixelated.
- Please address the queries from our data editors in the figure legends:
 1. Please define the annotated p values **** as well as provide the exact p-values for the same in the legends of figures EV7 B, D as appropriate.
 2. Please note that the exact p values are not provided in the legends of figures 1B, D; 2B, C, E, G, I; 3E, G; 4B, D, F, H; 5A, D; 6B, D, F; 8B, C, E, G; 9B, E, G, I; EV1 B, EV3; EV4 B, C, E, F;
 3. Please indicate the statistical test used for data analysis in the legends of figures EV7 B, D
 4. Please note that information related to n is missing in the legends of figures 7D, 8E, G, I, J; EV7 B, D
 5. Please note that the error bars are not defined in the legends of figures 7B, D; 9B, E, G, I; EV3, EV4 B, C, E, F; EV7B, D

3/ Thank you for providing Source Data. Please upload them as zip folders, 1 folder per figure. Please make sure all requested SD are provided. Please correct the nomenclature for the EV figures SD to Fig. EV1, etc.

4/ Checklist:

- Please check the section 'Cell materials'. If cell lines were used, please indicate whether they were authenticated and tested for mycoplasma contamination. Please update the Methods section of the manuscript text accordingly.
- Please fill in the subsection "Inclusion/exclusion criteria".

5/ Please update the Paper Explained according to the requested revisions, and include it in the main manuscript file.

6/ Thank you for providing the synopsis text. Please update and also provide a visual abstract to illustrate your article as a PNG file 550 px wide x 300-600 px high. A cropped portion of this image will serve as thumbnail for the table of content on our webpage.

7/ As part of the EMBO Publications transparent editorial process initiative (see our Editorial at <http://embomolmed.embopress.org/content/2/9/329>), EMBO Molecular Medicine will publish online a Review Process File (RPF) to accompany accepted manuscripts.

This file will be published in conjunction with your paper and will include the anonymous referee reports, your point-by-point

response and all pertinent correspondence relating to the manuscript. Let us know whether you agree with the publication of the RPF and as here, if you want to remove or not any figures from it prior to publication. Please note that the Authors checklist will be published at the end of the RPF.

I look forward to receiving your revised manuscript.

Yours sincerely,

Lise Roth

***** Reviewer's comments *****

Referee #1 (Remarks for Author):

The authors addressed most of the reviewers' comments. The manuscript is improved. The statin data is challenging to interpret per se; however, the overall interpretation of the data is supported by various approaches, and the limitations are better highlighted in the text. I agree with reviewer three that sometimes quantification of immunoblot can be useful; however, this is a semi-quantitative technique, and quantification can also be misleading. The ability to replicate the data in independent experiments and confirm some of the findings using quantifiable techniques such as ELISA and cell death assays is itself convincing.

Referee #3 (Comments on Novelty/Model System for Author):

1. Western Blot images were not quantified and consequently the differences were not statistically tested.
2. Concentration of simvastatin is 1000-fold higher than the pharmacologic dose and the concentration at which mechanism and specificity of its action was established.

Referee #3 (Remarks for Author):

The authors satisfactorily addressed several of my comments, but not all of them. Specifically, I've got problems with the rebuttals for comments #1 and #7.

1. I was not arguing against use of WB per se, however, the images must be quantified and statistics applied. I strongly disagree with the following statement "Although protein bands in Western blots can be quantified by densitometry, we believe this is here redundant since the differences between conditions are clearly distinguishable by eye." Statistics is obligatory for reporting scientific data, and there is a good reason for that, "distinguishing by eye" is so very often deceptive. Quantitation is also essential as it allows for analysis of associations. And I can't understand the reason for not doing this.
7. [REDACTED: referee comments and author response on unpublished data].

The authors addressed the editorial issues.

16th Jul 2025

Dear Prof. van Loo,

Thank you for submitting your revised files. Almost everything is fine now, however, before I can accept your manuscript, please address the following remaining issues:

1. Thank you for providing SD for the replicate WB experiments. Please add them to the SD files for each figure (i.e. replicate for Fig. 2A in the Figure 2A folder).
2. Please compile the WB figures directly from the original captured images to avoid image degradation. Although the provided figures are larger in dimensions, the resolution and quality of the blots have not improved. The provided source data is also at a resolution so small that it is unusable to validate the authenticity of the image set (please see examples attached of different figure sets under photoshop filters). Please let us know if you encounter any problem.
3. Thank you for providing a nice visual abstract. I have cropped a small portion to serve as thumbnail for the table of content on our webpage (attached). Please let me know if you agree, or provide an alternative image (115x70).

I look forward to receiving your revised manuscript.

Yours sincerely,

Lise Roth

The authors addressed the remaining editorial issues.

30th Jul 2025

Dear Prof. van Loo,

Thank you for submitting your revised files and uploading new high resolution figure files. I am pleased to inform you that your manuscript is accepted for publication and is now being sent to our publisher to be included in the next available issue of EMBO Molecular Medicine.

With kind regards,

Lise
